# Learning Multi-Agent Coordination via Sheaf-ADMM

**Jeffrey Seely** [* 1]   **Bartłomiej Cupiał** [* 1 2 3]   **Llion Jones** [1]

## Abstract

We present a differentiable optimization framework for multi-agent coordination. An input is decomposed into overlapping local views, each processed by an agent that solves a convex subproblem parameterized by a neural encoder. Agents coordinate through the Alternating Direction Method of Multipliers (ADMM) with inter-agent constraints specified by a cellular sheaf. The sheaf specifies which aspects of neighboring solutions must agree, allowing for heterogeneous notions of global consensus. Backpropagating through the unrolled optimization jointly trains all components of the multi-agent system. We evaluate on maze pathfinding, image classification, and Sudoku, where agents with individually insufficient local views learn to coordinate to produce correct global outputs. On MNIST, the local-view decomposition yields improved robustness to distribution shifts relative to a standard CNN. On Sudoku, the optimization-derived structure yields markedly higher solve rates than parameter-matched MPNN baselines. Finally, the ADMM structure exposes distinct primal, consensus, and dual state variables, opening the coordination dynamics to direct analysis and intervention—a property unavailable in standard message-passing architectures.

## 1. Introduction

Standard neural architectures are *monolithic*: a single large network processes inputs as a unified entity. By contrast, much of the intelligence observed in nature is *collective* (Ha & Tang, 2022; Sumpter, 2010), arising from groups of agents with limited local views that coordinate to solve global tasks. Inspired by this gap, we investigate a framework for learning multi-agent coordination on tasks where

---
[*]Equal contribution [1]Sakana AI, Tokyo, Japan [2]University of Warsaw [3]AKCES NCBR. Correspondence to: Jeffrey Seely <jeffrey@sakana.ai>.

*Proceedings of the 43rd International Conference on Machine Learning*, Seoul, South Korea. PMLR 306, 2026. Copyright 2026 by the author(s).

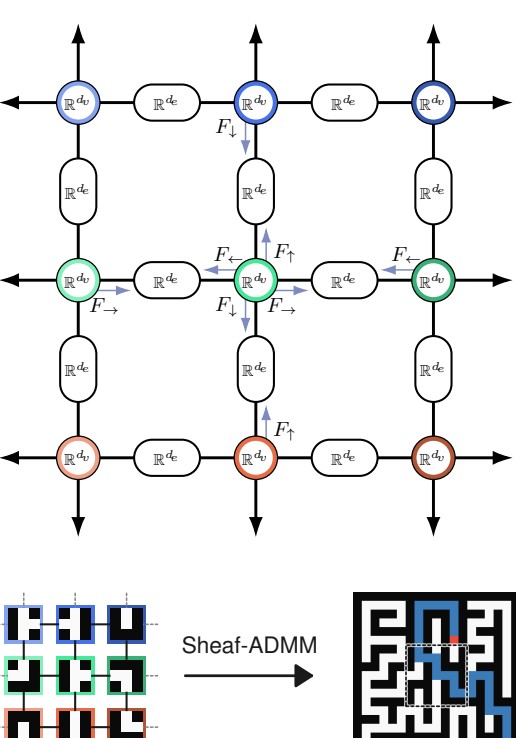

*Figure 1.* Each agent (circle) holds latent variables $\mathbf{x}_i, \mathbf{z}_i, \mathbf{u}_i \in \mathbb{R}^{d_v}$ (corresponding to primal, consensus, and dual variables, respectively), and selectively communicates with neighbors via linear maps $\mathbf{F}_{\text{direction}} \in \mathbb{R}^{d_e \times d_v}$ to reach consensus on shared edge spaces (ellipses). Sheaf-ADMM learns to coordinate such systems to solve tasks like maze pathfinding that require collective coordination.

each agent observes only a small local patch of an input. The method—Sheaf-ADMM—draws on two complementary fields: distributed optimization, which provides algorithms for decomposing large problems into local subproblems (Boyd et al., 2011), and cellular sheaf theory (Curry, 2014), which formalizes what it means for neighboring agents to agree on shared information (Hansen & Ghrist, 2021). The core idea is simple: each agent solves a local optimization problem parameterized by its view, then selectively communicates with neighbors, iterating until a global solution emerges (Figure 1).

Concretely, we formulate coordination as a constrained optimization problem and solve it using the Alternating Direc-

tion Method of Multipliers (ADMM). ADMM decomposes naturally into three steps per iteration: agents independently solve local subproblems (the **x**-update), a consensus step projects their proposals toward global consistency (the **z**-update), and dual variables accumulate the history of disagreement (the **u**-update).

A key design choice is how to define global consistency. A common choice asks agents to agree on their entire state vectors. Alternatively, using the language of *cellular sheaves* (Hansen & Ghrist, 2021; Hanks et al., 2025b), we specify that agents need only agree on projections of their states into shared edge spaces. Two agents solving adjacent regions in a maze pathfinding task need only coordinate on whether paths connect at their boundary, not on their entire internal structures.

In Sheaf-ADMM, individual agent subproblems are parameterized by a neural network encoder. The final iterate of ADMM is then processed by a neural network decoder. The entire pipeline is differentiable. We unroll a fixed number of ADMM iterations and backpropagate through the optimization trajectory (Monga et al., 2021).

This work extends prior work on utilizing ADMM in sheaf-constrained multi-agent systems (Hanks et al., 2025b), which used fixed sheaf structure for multi-agent linear control problems. Our contribution extends Hanks et al. (2025b) to a fully differentiable system and evaluates it in standard deep learning contexts. We evaluate Sheaf-ADMM on MNIST image classification and two reasoning tasks, maze pathfinding and Sudoku, in a regime where each agent observes only a small local patch that is insufficient to solve the task independently.

Sheaf-ADMM's iteration structure mirrors that of recurrent MPNNs (Gilmer et al., 2017; Li et al., 2016); other related architectures include sheaf neural networks (Bodnar et al., 2022) and neural cellular automata (Mordvintsev et al., 2020). Sheaf-ADMM differs from these architectures in two ways: (i) each agent maintains three distinct state variables (a local proposal, a consensus iterate, and a dual accumulator) rather than a single hidden vector, and (ii) both the per-agent and message-passing updates are constrained to optimization-derived forms (proximal maps and sheaf-Laplacian operations) rather than arbitrary learned nonlinear maps.

The combination of (i) and (ii) yields a framework that is interpretable and cleanly separates local computation from inter-agent coordination. The resulting inductive bias differs substantially from that of MPNN-style architectures, with distinct generalization and robustness behavior. Further, formulating Sheaf-ADMM in the languages of ADMM and sheaf theory lets us draw on the analytical and practical tools of both fields, yielding a rich toolkit for extensions.

## 2. Related Work

Our work draws on two threads: cellular sheaves as a structure for heterogeneous inter-agent communication (Curry, 2014), and distributed optimization via ADMM (Boyd et al., 2011).

Cellular sheaves generalize weighted graphs by assigning vector spaces to vertices and edges, together with linear restriction maps along incidences. This induces sheaf Laplacians that extend classical graph Laplacians and enable spectral and diffusion-based consistency mechanisms on heterogeneous network data (Hansen & Ghrist, 2019c;b). Sheaves have been used to model multi-agent dynamics including opinion formation (Hansen & Ghrist, 2021) and to impose global consistency constraints in optimization (Hansen & Ghrist, 2019a). In control and coordination, Hanks et al. (2025b) develop sheaf-constrained ADMM for multi-agent problems with fixed, hand-specified sheaves, with subsequent extensions to broader coordination settings (Zhao et al., 2025; Hanks et al., 2025a). Our framework retains the ADMM semantics from Hanks et al. (2025b) but learns the sheaf structure and local agent subproblem parameterizations end-to-end from data.

In graph representation learning, sheaves have also been used to define learnable diffusion operators that generalize GNN message passing; restriction maps are optimized from data to address heterophily and over-smoothing (Hansen & Gebhart, 2020; Bodnar et al., 2022).

Learning through unrolled optimization has a long history, from sparse-coding unrollings such as LISTA (Gregor & LeCun, 2010) to modern surveys (Monga et al., 2021). ADMM-Net (Yang et al., 2016) and follow-on work learn penalties, step sizes, and iteration schedules (Xie et al., 2019; Noah & Shlezinger, 2025; Saravanos et al., 2025). Differentiable optimization layers enable end-to-end training with embedded solvers via implicit differentiation (Amos & Kolter, 2017; Agrawal et al., 2019) and learned primal–dual updates (Adler & Öktem, 2018). Recent work connects distributed ADMM to message-passing GNNs (Doerks et al., 2026).

Finally, the iterative nature of ADMM connects to recurrent and fixed-point architectures for adaptive test-time compute (Wang et al., 2025b; Graves, 2016; Bai et al., 2019). Unlike generic recurrent blocks, our iterations correspond to primal proposals, consensus projections, and dual accumulation, enabling residual-based stopping.

## 3. Background

We briefly review the Alternating Direction Method of Multipliers (ADMM) and cellular sheaves, which form the algorithmic and structural foundations of our approach.

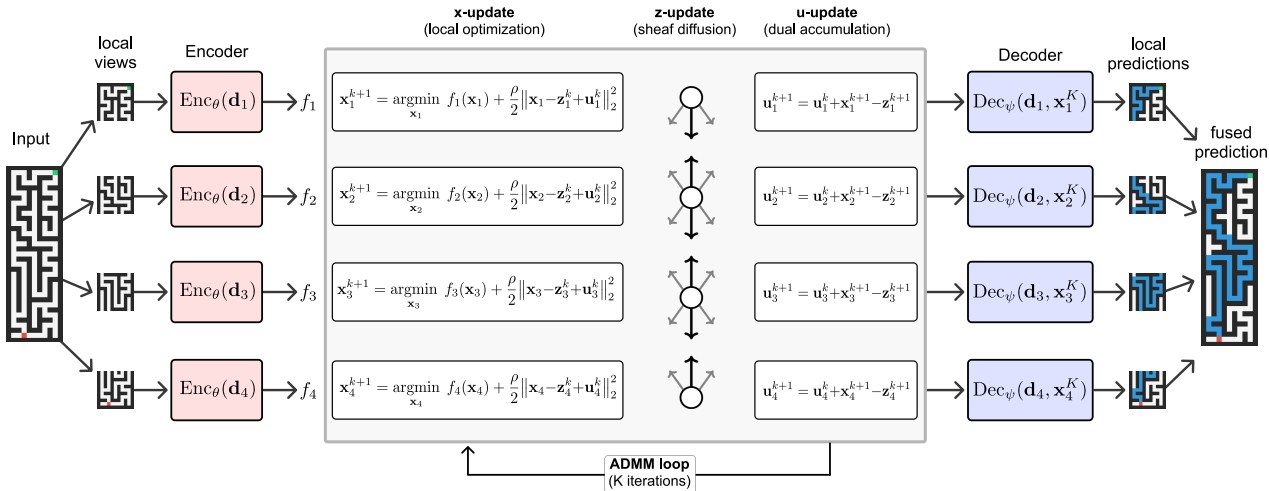

*Figure 2.* **The Sheaf-ADMM Architecture for the maze task.** Input is decomposed into local patches and processed by a shared encoder to produce optimization parameters (e.g., $\mathbf{Q}_i, \mathbf{q}_i$). These parameterize the ADMM layer, unrolled for $K$ iterations. Agents alternate between local optimization (**x**-update) and global coordination via sheaf diffusion (**z**-update), while dual variables **u** track disagreements. A decoder generates local predictions from final **x** and local patches. Local predictions are aggregated into the global output. The pipeline is fully differentiable, enabling end-to-end learning of coordination dynamics.

## 3.1. ADMM and Consensus Optimization

The Alternating Direction Method of Multipliers (ADMM) is a powerful framework for distributed optimization that decomposes large problems into smaller, parallelizable sub-problems (Boyd et al., 2011). Consider a problem of the form

$$\underset{\mathbf{x},\mathbf{z}}{\text{minimize}}\ f(\mathbf{x}) + g(\mathbf{z}) \quad \text{subject to} \quad \mathbf{x} = \mathbf{z} \quad (1)$$

where $\mathbf{x}, \mathbf{z} \in \mathbb{R}^d$ and $f, g$ are convex functions. ADMM solves this by forming the augmented Lagrangian

$$\mathcal{L}_\rho(\mathbf{x},\mathbf{z},\mathbf{u}) = f(\mathbf{x}) + g(\mathbf{z}) + \frac{\rho}{2}\|\mathbf{x} - \mathbf{z} + \mathbf{u}\|^2 \quad (2)$$

where $\mathbf{u} \in \mathbb{R}^d$ is the scaled dual variable and $\rho > 0$ is a penalty parameter. The algorithm alternates between minimizing over $\mathbf{x}$, minimizing over $\mathbf{z}$, and updating the dual variable $\mathbf{u}$:

$$\mathbf{x}^{k+1} = \underset{\mathbf{x}}{\text{argmin}}\ f(\mathbf{x}) + \frac{\rho}{2}\|\mathbf{x} - \mathbf{z}^k + \mathbf{u}^k\|^2 \quad (3)$$

$$\mathbf{z}^{k+1} = \underset{\mathbf{z}}{\text{argmin}}\ g(\mathbf{z}) + \frac{\rho}{2}\|\mathbf{x}^{k+1} - \mathbf{z} + \mathbf{u}^k\|^2 \quad (4)$$

$$\mathbf{u}^{k+1} = \mathbf{u}^k + \mathbf{x}^{k+1} - \mathbf{z}^{k+1} \quad (5)$$

The algorithm proceeds for $K$ iterations or until stopping criteria are met.

**Consensus form.** A particularly useful instantiation arises when $\mathbf{x}$ decomposes across $N$ agents, each with a local objective $f_i$. The *consensus* form of ADMM solves

$$\underset{\mathbf{x}}{\text{minimize}}\ \sum_{i=1}^{N} f_i(\mathbf{x}_i) \quad \text{subject to} \quad \mathbf{x} \in \mathcal{C} \quad (6)$$

where $\mathbf{x} = [\mathbf{x}_1; \ldots; \mathbf{x}_N]$ stacks the agent states and $\mathcal{C}$ encodes coupling constraints between agents. This maps onto (1) by setting $f(\mathbf{x}) = \sum_i f_i(\mathbf{x}_i)$ and $g(\mathbf{z}) = \chi_\mathcal{C}(\mathbf{z})$, the indicator function that is zero when $\mathbf{z} \in \mathcal{C}$ and infinite otherwise. The $\mathbf{z}$-update (4) then becomes Euclidean projection onto $\mathcal{C}$:

$$\mathbf{z}^{k+1} = \Pi_\mathcal{C}(\mathbf{x}^{k+1} + \mathbf{u}^k) \quad (7)$$

A natural choice for coupling constraints is full agreement: $\mathcal{C} = \{\mathbf{z} \mid \mathbf{z}_1 = \cdots = \mathbf{z}_N\}$. The projection (7) then reduces to computing the global mean, $\mathbf{z}_i^{k+1} = \frac{1}{N}\sum_j(\mathbf{x}_j^{k+1} + \mathbf{u}_j^k)$ for each $i$.

It is often desirable—or unavoidable—to compute (7) in a decentralized fashion. Given an agent graph, one can converge to the global mean via repeated pairwise averages between neighboring agents (requiring a connected graph and doubly stochastic mixing weights). When not iterated to convergence, this constitutes *undersolving* the $\mathbf{z}$-update, known as inexact ADMM.

**Interpretation.** At the first iterate $k = 0$, initializing $\mathbf{z}^0 = \mathbf{u}^0 = \mathbf{0}$, the $\mathbf{x}$-update corresponds to each agent proposing a locally greedy decision. The $\mathbf{z}$-update then finds the closest globally consistent configuration from each of those initial proposals, and $\mathbf{u}$ accumulates the discrepancy. At subsequent iterates, the augmented Lagrangian term $\|\mathbf{x} - \mathbf{z} + \mathbf{u}\|^2$ penalizes each agent's $\mathbf{x}$-update from straying too far from $\mathbf{z} - \mathbf{u}$: the consensus target, shifted by accumulated past error. Each agent thus carries three $d_v$-dimensional vectors: its local decision $\mathbf{x}_i$, the nearest consistent state $\mathbf{z}_i$, and an integral of accumulated error $\mathbf{u}_i$.

### 3.2. Cellular Sheaves

Full agreement in $\mathcal{C}$ can be too restrictive in certain tasks: not all agents need to agree on all components of their state. A more flexible notion of consensus allows neighboring agents to agree only on low-dimensional projections. For two connected agents $i, j$, linear maps $\mathbf{F}_i, \mathbf{F}_j$ can extract these relevant summaries, projecting high-dimensional agent states into a lower-dimensional agreement space. A cellular sheaf formalizes exactly this structure.

**Sheaf preliminaries.** Consider a graph $G = (\mathcal{V}, \mathcal{E})$ where each vertex $i \in \mathcal{V}$ corresponds to an agent with state $\mathbf{x}_i \in \mathbb{R}^{d_v}$, and edges encode which pairs of agents must coordinate. Each edge $e \in \mathcal{E}$ is assigned its own *edge stalk* $\mathbb{R}^{d_e}$—the space in which neighboring agents compare their states.

**Restriction maps.** For each edge $e = (i, j)$, linear maps $\mathbf{F}_{i \to e} \in \mathbb{R}^{d_e \times d_v}$ and $\mathbf{F}_{j \to e} \in \mathbb{R}^{d_e \times d_v}$ project agent states into the edge stalk. Global consistency requires $\mathbf{F}_{i \to e} \mathbf{x}_i = \mathbf{F}_{j \to e} \mathbf{x}_j$ for all edges. We now define a cellular sheaf $\mathcal{F}$ over $G$ as the collection of vector spaces assigned to each vertex and edge, and restriction maps to each vertex-edge incidence.

**Coboundary.** The sheaf generalization of the graph incidence matrix is the coboundary. Let $\mathbf{x} = [\mathbf{x}_1; \ldots; \mathbf{x}_N]$ stack all agent states. The restriction maps assemble into a block-sparse coboundary matrix $\mathbf{F}$ with one block-row per edge: for $e = (i, j)$, entries $\mathbf{F}_{i \to e}$ and $-\mathbf{F}_{j \to e}$ appear in column-blocks $i$ and $j$, zeros elsewhere. Then $\mathbf{Fx} = 0$ implies no disagreement at any edge. This yields a natural choice of constraint set: $\mathcal{C} = \{\mathbf{x} \mid \mathbf{x} \in \ker(\mathbf{F})\}$ for (6).

**Sheaf Laplacian and diffusion.** The sheaf Laplacian $\mathbf{L}_\mathcal{F} = \mathbf{F}^\top \mathbf{F}$ measures total disagreement:

$$\mathbf{x}^\top \mathbf{L}_\mathcal{F} \mathbf{x} = \sum_{e=(i,j)} \|\mathbf{F}_{i \to e} \mathbf{x}_i - \mathbf{F}_{j \to e} \mathbf{x}_j\|^2 \qquad (8)$$

Starting from an arbitrary initial state $\mathbf{x}^0$, gradient descent on $\mathbf{x}^\top \mathbf{L}_\mathcal{F} \mathbf{x}$ converges to the projection of $\mathbf{x}^0$ onto $\ker(\mathbf{L}_\mathcal{F}) = \ker(\mathbf{F})$:

$$\mathbf{x}^{t+1} = \mathbf{x}^t - \eta \mathbf{L}_\mathcal{F} \mathbf{x}^t \qquad (9)$$

This provides a method to compute the $\mathbf{z}$-update (7). Because $\mathbf{L}_\mathcal{F}$ is block-sparse, the update is local: each agent corrects its state based on disagreements with neighbors, weighted by the restriction maps. This *sheaf diffusion* generalizes pairwise averaging in decentralized consensus. To see this, note that if $G$ is fully connected and all restriction maps are identity, diffusion converges to the average: $\mathbf{x}_i^\infty = \frac{1}{N} \sum_j \mathbf{x}_j^0$ for each $i$.

**Conditioning.** We are interested in the regime with hundreds of agents and sparse connectivity. This drastically increases the condition number of $\mathbf{L}_\mathcal{F}$ and running diffusion to convergence is prohibitively expensive. Diffusion affects eigenmodes of $\mathbf{L}_\mathcal{F}$ at rates proportional to their eigenvalues: mode $i$ decays as $(1 - \eta \lambda_i)^t$. High-eigenvalue modes (local disagreements) decay rapidly; low-eigenvalue modes (global structure) decay slowly. Undersolving the $\mathbf{z}$-update—running only a few diffusion steps—thus acts as a smoother: it removes high-frequency disagreements while preserving coarse structure. This generically applies even in cases where $\ker(\mathbf{F}) = \mathbf{0}$, which may occur with a large number of edges compared to vertices, or with a large edge dimension.

## 4. Method

We present a differentiable optimization framework that coordinates multiple agents via sheaf-constrained ADMM, with all components learned end-to-end.

### 4.1. Problem Setup

Consider structured prediction tasks where an input $\mathbf{D} \in \mathbb{R}^{H \times W \times C_{\text{in}}}$ must be transformed into an output $\mathbf{Y} \in \mathbb{R}^{H \times W \times C_{\text{out}}}$.

We decompose the input into $N$ overlapping *views*. Each view $i$ corresponds to a patch $\mathbf{d}_i \in \mathbb{R}^{w \times w \times C_{\text{in}}}$ centered at position $(r_i, c_i)$ with receptive field $w$. Views are arranged on a grid with stride $s$. Each view defines an *agent*.

Agents are connected by a graph $G = (\mathcal{V}, \mathcal{E})$. For grid-structured inputs, we use 4-way or 8-way connectivity: each agent communicates with its spatial neighbors. With stride $s$ on an $H \times W$ input, this yields $N = \lceil H/s \rceil \cdot \lceil W/s \rceil$ agents.

The decomposition into views and the communication graph are independent design choices. The input need not be a grid, and the agent graph need not mirror the data structure *per se*. We focus on grid inputs with spatially local connectivity, but the framework applies to arbitrary graphs over arbitrary data subdivisions.

### 4.2. Overview

Each agent $i$ carries a decision variable $\mathbf{x}_i \in \mathbb{R}^{d_v}$. A shared encoder $\text{Enc}_\theta$ maps each agent's local view to parameters of a local convex objective $f_i$. Agents are related by learnable restriction maps. Agents then run $K$ iterations of ADMM toward solving (6). A shared decoder $\text{Dec}_\psi$ maps final states $\mathbf{x}_i^K$ to local predictions, which are averaged to form the global output (Figure 2). The entire pipeline is differentiable.

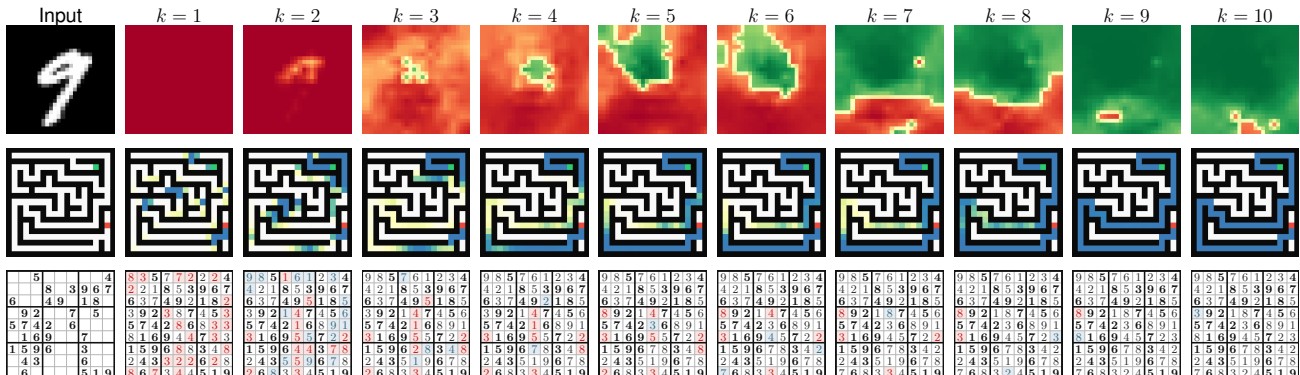

*Figure 3.* **Visualization of intermediate predictions by Sheaf-ADMM on benchmark tasks. Top:** MNIST—red/green pixels indicate the incorrect/correct predictions for each agent. **Middle:** Maze—blue cells indicate the predicted path. **Bottom:** Sudoku—bold digits represent initial givens; red highlights indicate cells currently violating Sudoku constraints; blue shading indicates updates from the previous timestep.

### 4.3. Local Objectives and the x-Update

Each agent's x-update solves the proximal subproblem from (3):

$$\mathbf{x}_i^{k+1} = \underset{\mathbf{x}_i}{\operatorname{argmin}} \; f_i(\mathbf{x}_i) + \frac{\rho}{2}\|\mathbf{x}_i - \mathbf{z}_i^k + \mathbf{u}_i^k\|^2 \qquad (10)$$

An encoder network $\text{Enc}_\theta(\mathbf{d}_i)$ produces the parameters of $f_i$, shared across agents. We focus on two primary choices of $f_i$, quadratic and quadratic with $\ell_1$ regularization:

**Quadratic.** With $f_i(\mathbf{x}_i) = \frac{1}{2}\mathbf{x}_i^\top \mathbf{Q}_i \mathbf{x}_i + \mathbf{q}_i^\top \mathbf{x}_i$ where $\mathbf{Q}_i \succeq 0$, the augmented Lagrangian absorbs into the quadratic:

$$\mathbf{x}_i^{k+1} = (\mathbf{Q}_i + \rho\mathbf{I})^{-1}\big(\rho(\mathbf{z}_i^k - \mathbf{u}_i^k) - \mathbf{q}_i\big) \qquad (11)$$

This is a single linear solve per agent. The encoder outputs $(\mathbf{Q}_i, \mathbf{q}_i)$. When $\mathbf{Q}_i$ is diagonal, this reduces to elementwise division. When $\mathbf{Q}_i$ is not diagonal, the inverse need only be computed once per forward pass.

**Diagonal quadratic + $\ell_1$.** Adding $\ell_1$ regularization $\lambda_i\|\mathbf{x}_i\|_1$ to $f_i$ admits a closed-form solution of (10) via soft-thresholding and clipping (Appendix A). The encoder outputs the diagonal of $\mathbf{Q}_i$, along with $\mathbf{q}_i$, and $\lambda_i$; softplus ensures positivity of $\mathbf{Q}_i$ and $\lambda_i$.

Further, for general QP subproblems, $f_i$ may include indicator functions for equality ($\mathbf{A}_i\mathbf{x}_i = \mathbf{b}_i$) and inequality ($\mathbf{G}_i\mathbf{x}_i \leq \mathbf{h}_i$) constraints. The subproblems are trainable via differentiable optimization layers (Amos & Kolter, 2017; Agrawal et al., 2019). Typically, each agent's state is low-dimensional, making these solves tractable relative to a global formulation. In this case, the encoder outputs $(\mathbf{Q}_i, \mathbf{q}_i, \mathbf{A}_i, \mathbf{b}_i, \mathbf{G}_i, \mathbf{h}_i)$. See Appendix A for details.

### 4.4. Restriction Maps

Restriction maps $\mathbf{F}_{i\to e}$ determine what information agents share. On grid graphs, edges have spatial directions (up, down, left, right for 4-connectivity). We learn a base restriction map per direction, shared across all agents.

The encoder outputs low-rank updates $\Delta\mathbf{F}_i = \mathbf{U}_i\mathbf{V}_i^\top$ that modulate the base maps: $\mathbf{F}_{i\to e} \leftarrow \mathbf{F}_{i\to e} + \Delta\mathbf{F}_i$. This allows restriction maps to depend on local content while keeping the number of encoder outputs manageable.

### 4.5. The z-Update

The z-update enforces inter-agent coordination by solving

$$\mathbf{z}^{k+1} = \underset{\mathbf{z}}{\operatorname{argmin}} \; g(\mathbf{z}) + \frac{\rho}{2}\|\mathbf{x}^{k+1} - \mathbf{z} + \mathbf{u}^k\|^2 \qquad (12)$$

where $g$ encodes coupling constraints. Setting $g = \chi_{\text{ker}(\mathbf{F})}$ (hard constraints) reduces this to projection onto $\text{ker}(\mathbf{F})$, computed via sheaf diffusion (Section 3.2). Setting $g(\mathbf{z}) = \frac{\gamma}{2}\mathbf{z}^\top \mathbf{L}_\mathcal{F}\mathbf{z}$ (soft constraints) penalizes disagreement without enforcing it exactly. On small agent graphs, the z-step can be computed exactly with either formulation. On large graphs with sparse connectivity, the conditioning of $\mathbf{L}_\mathcal{F}$ deteriorates and an exact solve becomes impractical. To undersolve the z-step (inexact ADMM), we use gradient descent with Nesterov momentum, running a fixed number $T$ of diffusion steps to approximate (12); each step is local and admits a message-passing interpretation (Appendix G).

Because the z-update is a PSD linear system, any solver for such systems can be substituted in. We also use conjugate gradient (CG), which converges faster than GD on ill-conditioned $\mathbf{L}_\mathcal{F}$ while remaining decentralization-friendly: only matrix-vector products with block entries of $\mathbf{L}_\mathcal{F}$ (local operations) plus global inner products.

**Algorithm 1** Sheaf-ADMM

Sheaf-ADMM assumes a choice of: (i) the local convex objective family $f_i(\cdot;\phi_i)$ which determines prox_f: quadratic, diagonal+$\ell_1$, or QP, and (ii) the data subdivision of $\mathbf{D}$ and agent graph $G$.

```
# Sheaf-ADMM forward pass
def forward(d, F, K, T):
    # Encode: local views -> optimization parameters
    for i in agents:
        params[i] = encoder(d[i])

    # Sheaf-ADMM
    z, u = zeros(), zeros()
    for k in range(K):
        # x-update: local optimization (parallel)
        for i in agents:
            x[i] = prox_f(z[i] - u[i], params[i], rho)
        # z-update: sheaf diffusion (T steps)
        z = x + u
        for t in range(T):
            z = z - eta * L_F(params) @ z
        # u-update: dual accumulation
        u = u + x - z

    # Decode: final states -> predictions
    for i in agents:
        y_hat[i] = decoder(d[i], x[i])
    return aggregate(y_hat)
```

### 4.6. Decoder

After $K$ ADMM iterations, a shared decoder maps each agent's final state and local view to a local prediction:

$$\hat{\mathbf{y}}_i = \text{Dec}_\psi(\mathbf{d}_i, \mathbf{x}_i^K) \tag{13}$$

The global output $\hat{\mathbf{Y}}$ is assembled by averaging overlapping regions.

All operations are differentiable. We unroll $K$ ADMM iterations and minimize a task loss between the global prediction $\hat{\mathbf{Y}}$ and the target $\mathbf{Y}$. Gradients flow through decoder, ADMM iterations, and encoder. Beyond encoder weights $\theta$ and decoder weights $\psi$, we learn the base restriction maps $\{\mathbf{F}_{i\to e}\}$ (shared by direction) and the penalty $\rho$ (positivity enforced via softplus). The method is summarized in Algorithm 1.

## 5. Experiments

We evaluate Sheaf-ADMM on three tasks: MNIST classification, maze pathfinding, and Sudoku. Each task tests the framework's ability to achieve global coordination using agents constrained to limited local views.

### 5.1. Benchmarks

**MNIST classification.** We treat image classification as a distributed consensus problem. A $28 \times 28$ image is divided into non-overlapping $3 \times 3$ patches, creating a grid of agents where each observes 9 pixels. This tests whether agents can aggregate insufficient local evidence into a correct global

prediction. A single agent seeing a vertical line cannot distinguish between a 1, 4, or 7; the system must communicate to reach a consensus.

**Maze pathfinding.** To evaluate long-range information propagation, we train agents to identify the path in randomly generated mazes. We generate 10,000 mazes of pixel size $19 \times 19$ corresponding to a $9 \times 9$ maze when viewed as a path graph. Each maze is generated by depth-first search, producing loop-free mazes. The small maze size in the training data is to limit the system's ability to learn long-range paths from training data alone. Each agent views a $3 \times 3$ portion of the pixel image; see Figure 7 for an example. The minimum path length is set to 18 pixels and scales as $1.5\times$ the width for our experiments in size generalization.

**Multi-Agent Sudoku.** We evaluate reasoning on non-spatial graphs using $9 \times 9$ Sudoku puzzles from the Ritvik19 dataset (Rastogi, 2024). In this formulation, agents correspond to constraint groups (rows, columns, and $3 \times 3$ boxes) rather than spatial patches, with edges connecting groups that share cells. This task tests the framework's flexibility on general graphs and its ability to handle discrete constraints that cannot be satisfied by simple averaging.

### 5.2. Recurrent MPNN Baselines

To isolate the effect of the ADMM-derived update rules, we compare against recurrent MPNN baselines that use the same agent graph and the same encoder/decoder architectures. Each MPNN replaces the ADMM layer with a learned message-passing iteration: messages are computed by a learned edge function, aggregated at each agent, and used to update the agent's hidden state via a learned recurrent block (Li et al., 2016).

We construct four variants along two axes. *Parameter-matched* (PM) variants use a larger hidden dimension to match Sheaf-ADMM's total parameter count; *communication-matched* (CM) variants use the same $d_v$, resulting in fewer parameters. Each is tested with both max and mean aggregation. See Appendix G for more details.

### 5.3. Experimental Setup

**Agent decomposition and encoding.** We decompose inputs into $N$ overlapping views, each processed by an agent. For MNIST and mazes, agents correspond to $3 \times 3$ patches tiled spatially over the image. In Multi-Agent Sudoku, we have 9 row agents, 9 column agents, and 9 box agents, with edges connecting agents that share cells.

In all tasks, the encoder network is a single-layer MLP whose output is a concatenation of parameters of its local objective $f_i$ as well as low-rank modulations for the restriction maps.

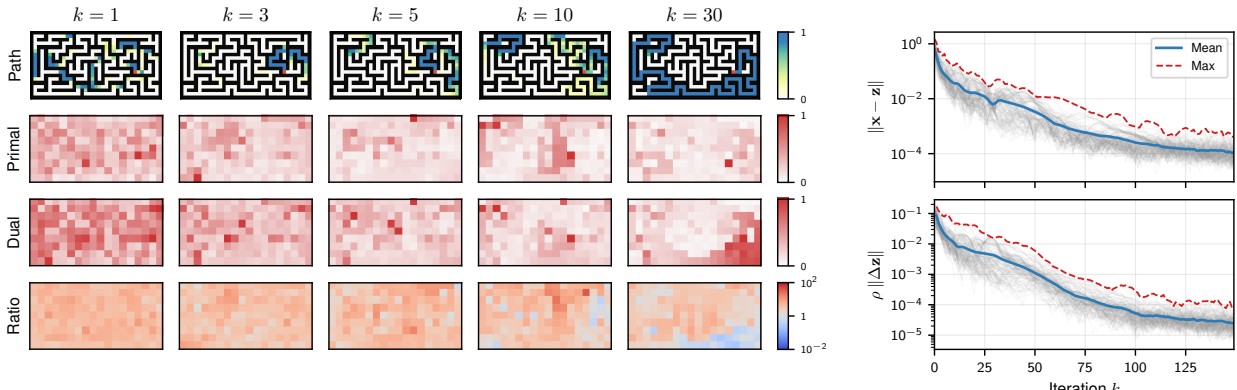

*Figure 4.* **Coordination dynamics on a** $2\times$ **out-of-distribution maze. Top row:** aggregated path prediction at iterations $k \in \{1, 3, 5, 10, 30\}$. **Middle rows:** per-agent primal residual $\|\mathbf{x}_i - \mathbf{z}_i\|$ and per-iteration dual residual $\rho\|\Delta\mathbf{z}_i\|$, each normalized as a fraction of the across-agent total per iteration. **Bottom row:** their log-ratio. **Right:** mean and max across agents of the same two quantities over iterations $k$. At late iterations, the dual residual is concentrated in the lower-right of the maze, near a branching point.

**ADMM layer.** Our typical setup uses a diagonal $\mathbf{Q}_i$ with $\ell_1$ regularization for the local $f_i$ objective in the $\mathbf{x}$-update. We use unrolled conjugate gradient (5 iterations) for the $\mathbf{z}$-update. Restriction maps have orthonormal initializations. Base restriction maps use task-specific weight sharing: global sharing for MNIST, directional for Maze, and slice-based sharing for Sudoku (9 shared base maps, each a selector onto one of 9 disjoint $d_e$-dimensional blocks of the vertex stalk).

A key hyperparameter choice is the vertex stalk dimension $d_v$ (the dimension of each $\mathbf{x}_i, \mathbf{z}_i, \mathbf{u}_i \in \mathbb{R}^{d_v}$) and the edge stalk dimension $d_e$ (each restriction map $\mathbf{F}_{i\to e} \in \mathbb{R}^{d_e \times d_v}$). In our default configuration, we set $(d_v, d_e)$ to $(32, 24)$ for MNIST, $(10, 5)$ for Maze, and $(288, 32)$ for Sudoku.

**Decoding and training.** The final $\mathbf{x}_i^K$ and local view $\mathbf{d}_i$ are passed to a shared decoder to output logits (per-pixel for mazes; per-cell for Sudoku), which are aggregated by averaging to form the global prediction. Models are trained end-to-end with cross-entropy loss.

### 5.4. Main Results

Sheaf-ADMM successfully coordinates agents across all three tasks. On Sudoku, Sheaf-ADMM achieves a 92.6% solve rate with 1.12M parameters, compared to 10.7% for the parameter-matched MPNN, and 34.7% for a 4.62M-parameter MPNN (Table 2). On Maze, Sheaf-ADMM matches the best MPNN on in-distribution test solve rate while using $d_v = 10$ vs. $d_v = 84$ (Table 1). We next ablate components of Sheaf-ADMM to understand which design choices drive these results.

**Sheaf structure.** We ablate the restriction map structure. Without coordination ($K = 0$), performance collapses to near-chance across all tasks. Fixed identity maps recover partial performance on MNIST but fail on Maze and Sudoku.

Learned shared maps suffice on Sudoku (92.5%) but not on Maze (8.9%); LoRA modulation closes the Maze gap (99.8%).

**Choice of $\mathbf{x}$- and $\mathbf{z}$-updates.** Table 3 ablates the choice of local objective $f_i$ and $\mathbf{z}$-update method. For the $\mathbf{x}$-update, adding $\ell_1$ regularization to the diagonal quadratic objective is critical on Maze and Sudoku, where the resulting soft-thresholding produces sparse local proposals. On MNIST, all $\mathbf{x}$-update variants perform comparably. For the $\mathbf{z}$-update, conjugate gradient substantially outperforms gradient descent on MNIST and Sudoku, consistent with the poor conditioning of $\mathbf{L}_{\mathcal{F}}$ on these graphs.

**Number of iterations.** Performance improves with the number of ADMM iterations $K$ up to a task-dependent saturation point (Table 3). MNIST saturates early ($K{=}10$), while Maze and Sudoku require $K{=}20$–$30$. Beyond saturation, performance can degrade, likely due to gradient degradation in deep unrolls.

**Qualitative evolution.** Figure 3 visualizes the emergence of global consensus over iterations. In maze pathfinding, the shortest path progressively emerges from locally plausible alternatives. In Sudoku, constraint violations are systematically resolved. On MNIST, agents initially disagree but converge over iterations to a per-pixel class consensus (Figure 5); the early disagreement patterns are consistent with local ambiguity between digits that share similar patches (e.g., 3 and 8, 7 and 9).

### 5.5. Out-of-Distribution Generalization

Because the encoder and decoder operate on local views with shared weights, Sheaf-ADMM can be applied to inputs that differ in size or structure from the training data without architectural changes.

*Table 1.* Maze recurrent-baseline sweep: exact solve rate (%), 3-seed mean ± std. Recurrent MPNN baselines are grouped into parameter-matched (PM) and communication-matched (CM) variants, each with max or mean aggregation.

| Model | $d_v$ | Params | Test | 2× OOD | 4× OOD |
|---|---|---|---|---|---|
| **Sheaf-ADMM** | 10 | 182K | **99.9**$_{\pm 0.1}$ | **98.1**$_{\pm 1.2}$ | **4.5**$_{\pm 1.1}$ |
| MPNN (PM; max) | 84 | 182K | **99.9**$_{\pm 0.1}$ | 68.3$_{\pm 2.4}$ | 1.3$_{\pm 1.0}$ |
| MPNN (PM; mean) | 84 | 182K | 93.8$_{\pm 6.3}$ | 52.0$_{\pm 25.4}$ | 0.5$_{\pm 0.8}$ |
| MPNN (CM; max) | 10 | 49K | 35.1$_{\pm 37.2}$ | 6.5$_{\pm 9.2}$ | 0.0$_{\pm 0.1}$ |
| MPNN (CM; mean) | 10 | 49K | 94.7$_{\pm 4.5}$ | 43.0$_{\pm 7.6}$ | 0.6$_{\pm 0.7}$ |

*Table 2.* Sudoku recurrent-baseline sweep on the test set (3-seed mean ± std). MPNN baselines are grouped into three configurations: parameter-matched to Sheaf-ADMM (fixed RM) at 1.15M (PM-fixed), communication-matched at $d_v$=288 (CM), and parameter-matched to Sheaf-ADMM (LoRA) at 4.62M (PM-LoRA). Each is tested with max or mean aggregation.

| Model | $d_v$ | Params | Solved | Cell acc |
|---|---|---|---|---|
| **Sheaf-ADMM (fixed RM)** | 288 | 1.12M | **92.6**$_{\pm 0.2}$ | **99.5**$_{\pm 0.0}$ |
| Sheaf-ADMM (LoRA) | 288 | 4.45M | 87.4$_{\pm 1.0}$ | 99.2$_{\pm 0.1}$ |
| MPNN (PM-fixed; max) | 225 | 1.15M | 10.7$_{\pm 0.6}$ | 86.3$_{\pm 0.7}$ |
| MPNN (PM-fixed; mean) | 225 | 1.15M | 4.0$_{\pm 0.2}$ | 74.8$_{\pm 1.5}$ |
| MPNN (CM; max) | 288 | 1.72M | 19.9$_{\pm 3.5}$ | 91.0$_{\pm 1.3}$ |
| MPNN (CM; mean) | 288 | 1.72M | 6.2$_{\pm 1.1}$ | 79.4$_{\pm 0.9}$ |
| MPNN (PM-LoRA; max) | 504 | 4.62M | 34.7$_{\pm 5.8}$ | 94.1$_{\pm 0.7}$ |
| MPNN (PM-LoRA; mean) | 504 | 4.62M | 10.8$_{\pm 1.2}$ | 85.3$_{\pm 1.1}$ |

**MNIST robustness.** Table 4 compares Sheaf-ADMM to a CNN baseline under distribution shifts: padding (larger canvas), patch dropout (missing agents), and Gaussian noise. Sheaf-ADMM is substantially more robust across all conditions. With 16-pixel padding, the CNN drops to 11.4% while Sheaf-ADMM retains 86.3%. With 30% patch dropout, Sheaf-ADMM achieves 69.1% vs. 45.5% for the CNN.

**Maze size generalization.** Models trained exclusively on $19 \times 19$ mazes generalize to larger inputs at test time. As shown in Figure 6, performance remains near-saturated up to approximately a 2× increase in linear resolution ($39 \times 39$ mazes). The aggregated path-belief heatmaps show that early iterations ($K \leq 2$) contain only fragmented, locally plausible evidence. As $K$ increases, the sheaf constraints prune globally inconsistent branches; by $K$=60, the prediction resolves to a single coherent shortest path on a $37 \times 37$ maze.

### 5.6. Additional Analysis

**Per-agent coordination dynamics.** Figure 4 visualizes per-agent primal residual $\|x_i - z_i\|$ and dual residual $\rho\|\Delta z_i\|$ over ADMM iterations. At late iterations, the dual residual concentrates near a branching point in the lower-right of the maze. The across-agent mean and max of each quantity (right) decrease over iterations.

*Table 3.* Combined ablations across tasks. We report Test Accuracy for MNIST and Test Solved Rate (%) for Maze and Sudoku.

| Ablation | Variant | MNIST (Acc) | Maze (Solved) | Sudoku (Solved) |
|---|---|---|---|---|
| Sheaf | No coordination ($K = 0$) | 11.0% | 0% | 0.0% |
| | Identity maps ($\mathbf{F}=\mathbf{I}$) | 64.2% | 0% | 6.1% |
| | Learned Shared Maps | 82.4% | 8.9% | **92.5%** |
| | Shared + LoRA | **98.5%** | **99.8%** | 87.5% |
| x-update | Diagonal (fixed $\mathbf{Q}$) | **98.4%** | 38.9% | 15.8% |
| | Diagonal quad | 98.2% | 99.2% | 47.0% |
| | Diagonal quad + $\ell_1$ | **98.5%** | **99.8%** | **92.5%** |
| z-update | GD + Nesterov | 52.2% | 98.2% | 0.0% |
| | Conjugate Gradient | **98.5%** | **99.8%** | **92.5%** |
| Iterations | $K = 1$ | 11.0% | 0% | 5.5% |
| | $K = 2$ | 19.0% | 0% | 13.6% |
| | $K = 5$ | 95.4% | 1.2% | 64.0% |
| | $K = 10$ | **98.5%** | 77.4% | 83.6% |
| | $K = 20$ | 97.7% | 86.6% | **92.5%** |
| | $K = 30$ | 90.3% | **99.8%** | 92.3% |

*Table 4.* Robustness evaluation on MNIST under distribution shifts. We compare a standard CNN baseline against Sheaf-ADMM (stride $s = 3$). **Padding:** The input image is zero-padded. **Patch Dropout:** Random patches (agents) are masked out during inference. **Input Noise:** Additive Gaussian noise.

| Experiment | Condition | CNN | Sheaf-ADMM |
|---|---|---|---|
| Baseline | - | **99.3%** | 98.5% |
| Padding | Pad 4 | 70.8% | **98.4%** |
| | Pad 8 | 44.1% | **97.5%** |
| | Pad 16 | 11.4% | **86.3%** |
| Patch Dropout | Drop 5% | 94.4% | **97.0%** |
| | Drop 10% | 84.0% | **94.6%** |
| | Drop 30% | 45.5% | **69.1%** |
| Gaussian Noise | $\sigma = 0.10$ | 54.0% | **74.0%** |
| | $\sigma = 0.20$ | 9.9% | **31.9%** |

**Local-vs-consensus trajectories.** Figure 8 in Appendix F shows, for each agent, the trajectories of the local proposal $\mathbf{x}_i^k$ (blue) and consensus variable $\mathbf{z}_i^k$ (red) across ADMM iterations, arranged in a grid matching agent spatial positions. Agents along straight corridors show short, nearly linear trajectories with tight overlap between local and consensus; agents at branching or decision points exhibit extended, non-linear trajectories where local proposals repeatedly revise against the consensus before converging.

## 6. Conclusion

### 6.1. Connections to the Broader ADMM Toolbox

Our current implementation explores a narrow slice of the ADMM design space. There is a large literature on practical enhancements. Examples include: over-relaxation and damping (Ghadimi et al., 2015), adaptive penalty selection, including spectral and residual-balancing heuristics (Xu et al., 2017; Wohlberg, 2017), preconditioning and scaling to improve conditioning of linear solves (Giselsson & Boyd,

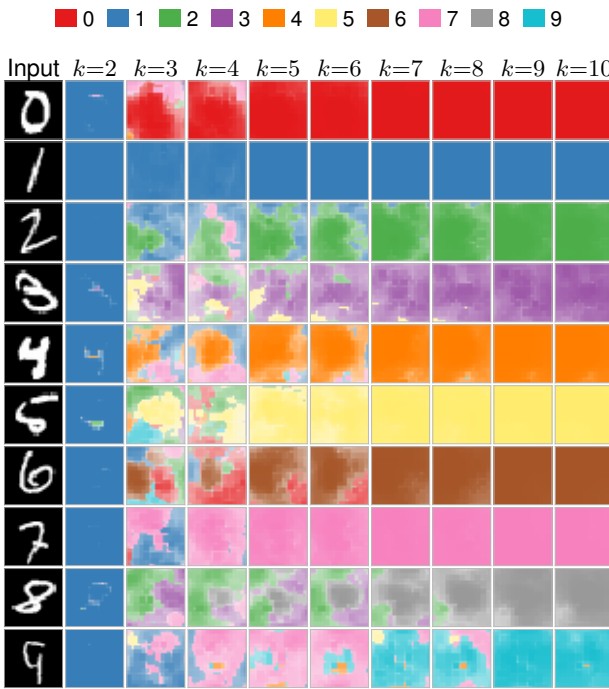

*Figure 5.* **Class emergence.** Pixel = argmax class across overlapping agents; intensity = confidence.

2014), and others (Yang et al., 2022).

### 6.2. Relation to Differentiable Optimization Layers and "Distributed OptNet"

Differentiable optimization layers embed a convex program inside a network and backpropagate through its solution, either via implicit differentiation of the KKT conditions or by unrolling an iterative solver (Amos & Kolter, 2017; Agrawal et al., 2019; Monga et al., 2021). These approaches are most compelling when the embedded problem is moderate-sized; in contrast, monolithic programs whose variable count grows with input size can be dominated by solving large KKT systems. Sheaf-ADMM can be read as a *distributed* differentiable optimization layer: we learn a decomposition into many small local convex subproblems coupled by sparse linear constraints, so inference is implemented by parallel local proximal solves plus sparse message passing. Closest in spirit are differentiable operator-splitting/ADMM layers and learned distributed QP solvers that also unroll ADMM-style updates (Xie et al., 2019; Noah & Shlezinger, 2025; Saravanos et al., 2025); our distinctive ingredient is that the coupling structure itself is learned via a cellular sheaf (restriction maps).

### 6.3. Limitations

The central inductive bias of our approach is that the task decomposes into overlapping local subproblems whose compatibility can be enforced through local constraints/messages. This assumption is not universally appropriate. Failures can occur when (i) the correct global prediction depends on nonlocal statistics that cannot be represented as low-dimensional overlap variables, (ii) the chosen agent graph does not reflect the true dependency structure (too few or misaligned overlaps), (iii) the learned restriction maps become effectively dense/high-dimensional potentially removing any benefit of sparse communication, or (iv) the problem requires long-range coordination but the unrolled horizon $K$ and choice of diffusion steps $T$ are too small.

ADMM is most frequently associated with convex optimization problems. In Sheaf-ADMM, the encoder and decoder handle non-convexity but the latent iterations are convex. Convexity allows for convergence guarantees but is not a strict requirement (Hong et al., 2016; Wang et al., 2019).

### 6.4. Future Directions

We suggest several directions we believe merit investigation: *asynchronous updates*, where agents update at different rates (Zhao et al., 2025); *cooperation and competition*, extending beyond consensus to mixed or adversarial incentives; incorporating advances from *sheaf neural networks* (e.g., Ribeiro et al. (2026); Fiorini et al. (2026)); *alternative partitioning* strategies such as augmented views, hierarchical decompositions, or multi-scale agents; and *dynamic environments* requiring coordination over time (Wang et al., 2025a). Beyond these practical directions, part of our motivation is to explore collective intelligence as a computational paradigm—understanding how groups of simple agents with limited views can coordinate to solve problems beyond any individual's capacity.

## Impact Statement

This paper presents work whose goal is to advance the field of machine learning. There are many potential societal consequences of our work, none of which we feel must be specifically highlighted here.

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

## A. Solvers for the x-Update

The x-update solves the proximal subproblem:

$$\mathbf{x}_i^{k+1} = \underset{\mathbf{x}_i}{\operatorname{argmin}} \; f_i(\mathbf{x}_i) + \frac{\rho}{2}\|\mathbf{x}_i - \mathbf{v}_i\|^2 \qquad (14)$$

where $\mathbf{v}_i = \mathbf{z}_i^k - \mathbf{u}_i^k$. The function $f_i$ is convex and may include indicator functions $\chi_{\mathcal{C}_i}$ to encode hard constraints. The choice of $f_i$ determines whether the subproblem admits a closed-form solution or requires an iterative solver.

### A.1. Quadratic (Closed Form)

With $f_i(\mathbf{x}) = \frac{1}{2}\mathbf{x}^\top \mathbf{Q}_i \mathbf{x} + \mathbf{q}_i^\top \mathbf{x}$ and $\mathbf{Q}_i \succeq 0$:

$$\mathbf{x}_i^{k+1} = (\mathbf{Q}_i + \rho\mathbf{I})^{-1}(\rho\mathbf{v}_i - \mathbf{q}_i) \qquad (15)$$

When $\mathbf{Q}_i$ is diagonal, this becomes elementwise: $x_j = (\rho v_j - q_j)/(Q_{jj} + \rho)$.

### A.2. Diagonal + $\ell_1$ + Box

With diagonal $\mathbf{Q}_i$, adding $\ell_1$ regularization $\lambda_1\|\mathbf{x}\|_1$, $\ell_2$ regularization $\frac{\lambda_2}{2}\|\mathbf{x}\|^2$, and box constraints $\underline{\mathbf{x}} \le \mathbf{x} \le \bar{\mathbf{x}}$ still admits a closed-form solution. Define:

$$a_j = Q_{jj} + \lambda_2 + \rho, \quad t_j = \frac{\rho v_j - q_j}{a_j}, \quad \kappa_j = \frac{\lambda_1}{a_j} \quad (16)$$

Then the solution is:

$$x_j^* = \text{clip}\big(\text{soft}(t_j, \kappa_j),\ \underline{x}_j,\ \bar{x}_j\big) \qquad (17)$$

where $\text{soft}(v, \tau) = \text{sign}(v) \max(|v| - \tau, 0)$ is soft-thresholding. In our methods, we incorporate only the $\ell_1$ term but box constraints are a cheap addition to incorporate inequality constraints without requiring a general QP solve.

### A.3. Equality-Constrained QP (Linear Solve)

Adding linear equality constraints $\mathbf{Ax} = \mathbf{b}$ to the quadratic case yields the KKT system:

$$\begin{bmatrix} \mathbf{Q}_i + \rho \mathbf{I} & \mathbf{A}^\top \\ \mathbf{A} & 0 \end{bmatrix} \begin{bmatrix} \mathbf{x} \\ \boldsymbol{\nu} \end{bmatrix} = \begin{bmatrix} \rho \mathbf{v}_i - \mathbf{q}_i \\ \mathbf{b} \end{bmatrix} \qquad (18)$$

With diagonal $\mathbf{Q}_i$, the Schur complement $\mathbf{A}(\mathbf{Q}_i + \rho \mathbf{I})^{-1} \mathbf{A}^\top$ is cheap to form, reducing to an $m \times m$ solve where $m$ is the number of constraints.

### A.4. General QP (Iterative)

With inequality constraints $\mathbf{Gx} \leq \mathbf{h}$ in addition to equalities and bounds, the subproblem becomes a general QP requiring an iterative solver (active-set, interior-point, or operator-splitting methods). Because each agent's subproblem is low-dimensional, these solves remain tractable.

## B. Solvers for the z-Update

The $\mathbf{z}$-update projects $\mathbf{v} = \mathbf{x}^{k+1} + \mathbf{u}^k$ onto $\ker(\mathbf{F})$:

$$\mathbf{z}^* = \underset{\mathbf{z}}{\arg\min} \frac{1}{2} \|\mathbf{z} - \mathbf{v}\|^2 \quad \text{subject to} \quad \mathbf{Fz} = 0 \qquad (19)$$

where $\mathbf{F}$ is the coboundary operator mapping vertex states to edge residuals. This is the orthogonal projection of $\mathbf{v}$ onto $\ker(\mathbf{F}) = \ker(\mathbf{L}_\mathcal{F})$, where $\mathbf{L}_\mathcal{F} = \mathbf{F}^\top \mathbf{F}$ is the sheaf Laplacian.

Iterative solvers for this projection may appear costly: each ADMM iteration requires solving the projection anew. However, the edge-space formulation admits warm-starting, and in practice one solves to a tolerance (absolute or relative) rather than running a fixed number of steps.

### B.1. Node-Space Solver

Gradient descent on the sheaf energy $\frac{1}{2}\mathbf{z}^\top \mathbf{L}_\mathcal{F} \mathbf{z}$ initialized at $\mathbf{z}^{(0)} = \mathbf{v}$ converges to the projection:

$$\mathbf{z}^{(t+1)} = \mathbf{z}^{(t)} - \eta\, \mathbf{L}_\mathcal{F} \mathbf{z}^{(t)} = \mathbf{z}^{(t)} - \eta\, \mathbf{F}^\top \mathbf{Fz}^{(t)} \qquad (20)$$

The matvec $\mathbf{L}_\mathcal{F} \mathbf{z}$ decomposes into local operations: for each edge $e = (i, j)$, compute the disagreement $\mathbf{F}_{i \to e} \mathbf{z}_i - \mathbf{F}_{j \to e} \mathbf{z}_j$, then scatter back to vertices via $\mathbf{F}^\top$. Warm-starting from $\mathbf{z}^{k-1}$ directly fails (it lies in $\ker(\mathbf{L}_\mathcal{F})$, so the gradient is zero), but initializing at $\mathbf{z}^{k-1} + (\mathbf{v}^k - \mathbf{v}^{k-1})$ works since projection is linear.

### B.2. Edge-Space Solver

Alternatively, solve in the edge (dual) space. The projection can be written as $\mathbf{z}^* = \mathbf{v} - \mathbf{F}^\top \boldsymbol{\lambda}^*$ where $\boldsymbol{\lambda}^* \in \mathbb{R}^{E \cdot d_e}$ (with $E := |\mathcal{E}|$) solves:

$$\mathbf{FF}^\top \boldsymbol{\lambda}^* = \mathbf{Fv} \qquad (21)$$

Gradient descent on $\frac{1}{2} \boldsymbol{\lambda}^\top \mathbf{FF}^\top \boldsymbol{\lambda} - \boldsymbol{\lambda}^\top \mathbf{Fv}$ gives:

$$\boldsymbol{\lambda}^{(t+1)} = \boldsymbol{\lambda}^{(t)} - \eta \left( \mathbf{FF}^\top \boldsymbol{\lambda}^{(t)} - \mathbf{Fv} \right) \qquad (22)$$

This is advantageous when $E \cdot d_e < N \cdot d_v$ (low-dimensional edge stalks). Crucially, warm-starting works here: $\mathbf{FF}^\top$ is fixed across ADMM iterations and only the right-hand side $\mathbf{Fv}$ changes, so $\boldsymbol{\lambda}$ can be initialized from the previous iteration's solution.

### B.3. Dual Interpretation

The edge-space formulation arises naturally from the Lagrangian dual. Introducing multipliers $\boldsymbol{\lambda}$ for the constraint $\mathbf{Fz} = 0$:

$$\mathcal{L}(\mathbf{z}, \boldsymbol{\lambda}) = \frac{1}{2} \|\mathbf{z} - \mathbf{v}\|^2 + \boldsymbol{\lambda}^\top \mathbf{Fz} \qquad (23)$$

Setting $\nabla_\mathbf{z} \mathcal{L} = 0$ gives $\mathbf{z} = \mathbf{v} - \mathbf{F}^\top \boldsymbol{\lambda}$. Substituting into the constraint yields $\mathbf{FF}^\top \boldsymbol{\lambda} = \mathbf{Fv}$.

### B.4. Soft Constraints

Recall the ADMM formulation minimizes $f(\mathbf{x}) + g(\mathbf{z})$ subject to $\mathbf{x} = \mathbf{z}$, where $g(\mathbf{z}) = \chi_{\ker(\mathbf{F})}(\mathbf{z})$ is the indicator enforcing $\mathbf{Fz} = 0$. The soft variant replaces this indicator with a quadratic penalty $g(\mathbf{z}) = \frac{\gamma}{2} \|\mathbf{Fz}\|^2$. The $\mathbf{z}$-update then becomes:

$$\mathbf{z}^* = \underset{\mathbf{z}}{\arg\min} \ \frac{\gamma}{2} \|\mathbf{Fz}\|^2 + \frac{\rho}{2} \|\mathbf{z} - \mathbf{v}\|^2 \qquad (24)$$

Setting the gradient to zero yields $(\rho \mathbf{I} + \gamma \mathbf{L}_\mathcal{F}) \mathbf{z} = \rho \mathbf{v}$, so:

$$\mathbf{z}^* = \rho(\rho \mathbf{I} + \gamma \mathbf{L}_\mathcal{F})^{-1} \mathbf{v} \qquad (25)$$

This is a "soft projection": it pulls toward consensus without enforcing it exactly. The ratio $\gamma/\rho$ controls the tradeoff—large $\gamma$ approximates the hard projection.

In edge space, the corresponding system is:

$$\left( \mathbf{I} + \frac{\gamma}{\rho} \mathbf{FF}^\top \right) \boldsymbol{\lambda}^* = \mathbf{Fv} \qquad (26)$$

The identity term ensures full rank and good conditioning, in contrast to the singular $\mathbf{FF}^\top$ in the hard-constraint case.

### B.5. Practical Considerations

Both undersolving (few GD steps on the hard problem) and soft constraints (converged solve with finite $\gamma$) produce

smoothed outputs rather than exact projections. The soft formulation guarantees a well-conditioned system in both node and edge space (due to the identity term), whereas the hard projection can be ill-conditioned in node space when $\mathbf{L}_{\mathcal{F}}$ has small nonzero eigenvalues. The edge-space hard system $\mathbf{FF}^{\top}$ may already be well-conditioned if $\mathbf{F}$ has full row rank. The choice between formulations is problem-dependent.

## C. Extensions

We highlight various extensions to the framework.

### C.1. Over-Relaxation

Over-relaxation replaces $\mathbf{x}^{k+1}$ with $\tilde{\mathbf{x}}^{k+1} = \alpha \mathbf{x}^{k+1} + (1 - \alpha)\mathbf{z}^k$ in the $\mathbf{z}$- and $\mathbf{u}$-updates. With $\alpha \in (1, 2)$, this can accelerate convergence by extrapolating past the current iterate. The value $\alpha = 1.5$–$1.8$ is typical. Over-relaxation is particularly effective when the $\mathbf{z}$-update is undersolved, as it compensates for incomplete convergence toward consensus.

### C.2. Residual Balancing

The penalty $\rho$ controls the tradeoff between primal feasibility ($\mathbf{x} \approx \mathbf{z}$) and dual convergence. A classical heuristic adjusts $\rho$ to balance residual magnitudes (Boyd et al., 2011):

$$\rho^{k+1} = \begin{cases} \tau \rho^k & \text{if } \|\mathbf{r}^k\| > \mu \|\mathbf{s}^k\| \\ \rho^k/\tau & \text{if } \|\mathbf{s}^k\| > \mu \|\mathbf{r}^k\| \\ \rho^k & \text{otherwise} \end{cases} \tag{27}$$

where $\mathbf{r}^k = \mathbf{x}^k - \mathbf{z}^k$ is the primal residual, $\mathbf{s}^k = \rho(\mathbf{z}^k - \mathbf{z}^{k-1})$ is the dual residual, and typical values are $\tau = 2$, $\mu = 10$. When $\rho$ changes, the dual variable should be rescaled: $\mathbf{u} \leftarrow \mathbf{u} \cdot \rho^k/\rho^{k+1}$.

### C.3. Caching Factorizations

When $\mathbf{Q}_i$ and $\rho$ are fixed across ADMM iterations, the matrix $\mathbf{Q}_i + \rho \mathbf{I}$ in the $\mathbf{x}$-update can be factored once (e.g., Cholesky) and reused. Similarly, in the soft-constraint $\mathbf{z}$-update, $\rho \mathbf{I} + \gamma \mathbf{L}_{\mathcal{F}}$ is fixed. This reduces per-iteration cost from $O(d^3)$ to $O(d^2)$ for the linear solves.

### C.4. Per-Agent Penalties

The penalty $\rho$ can vary across agents: $\rho_i$ for agent $i$. This allows different agents to have different consensus-vs-local tradeoffs, which may be useful when agents have heterogeneous objectives or operate at different scales. The $\mathbf{x}$- and $\mathbf{u}$-updates remain local; only the $\mathbf{z}$-update requires care to handle the weighted projection.

### C.5. Nonlinear Edge Potentials

Our framework uses either hard sheaf constraints ($\mathbf{Fz} = 0$) or quadratic soft penalties ($\frac{\gamma}{2}\|\mathbf{Fz}\|^2$). Hanks et al. (2025b) utilize a more general formulation using *nonlinear edge potentials* $\{U_e : \mathcal{F}(e) \to \mathbb{R}\}_{e \in E}$. The constraint becomes $\mathbf{L}_{\mathcal{F}}^{\nabla U} \mathbf{x} = 0$, where the *nonlinear sheaf Laplacian* is defined as:

$$\mathbf{L}_{\mathcal{F}}^{\nabla U} = \mathbf{F}^{\top} \circ \nabla U \circ \mathbf{F} \tag{28}$$

with $\nabla U$ acting coordinatewise on edge residuals. When $U_e(\mathbf{r}) = \frac{1}{2}\|\mathbf{r}\|^2$, this recovers the linear sheaf Laplacian $\mathbf{L}_{\mathcal{F}} = \mathbf{F}^{\top}\mathbf{F}$.

General edge potentials enable richer coordination semantics; for instance, Huber potentials tolerate outlier disagreements and $\ell_1$-like potentials allow some edges to disagree entirely.

## D. Size Generalization

**Setup.** We train the maze model exclusively on $19 \times 19$ mazes and evaluate zero-shot on larger $n \times n$ mazes, with $n \in \{19, 23, \ldots, 73\}$. At test time we run the learned Sheaf-ADMM updates for up to $K_{\max} = 100$ iterations. We report (i) *solved rate*, where a maze is counted as solved if the predicted path exactly matches the ground-truth shortest path, and (ii) the *first-solve iteration*—the smallest $K$ at which the model's prediction becomes correct (computed only for solved instances).

**Results.** Figure 6 shows that the same trained model remains highly effective up to roughly a $2\times$ increase in linear resolution. Qualitatively (top row), early iterates ($K \leq 2$) reflect only local evidence and contain many disconnected, locally plausible fragments. As $K$ increases, the sheaf constraints progressively prune inconsistent branches and sharpen a single globally coherent solution; by $K = 60$ the prediction is essentially fully resolved on a $37 \times 37$ maze. Quantitatively (bottom left), performance is near-saturated through $39 \times 39$ and then degrades as $n$ increases further, dropping sharply once the maze diameter exceeds what can be coordinated within the fixed iteration budget. The compute required to solve also grows with size (bottom right): the mean number of iterations to solve increases substantially with $n$, approaching the $K_{\max} = 100$ cap for the largest instances that are still solvable.

## E. Extended Ablation Studies

**Impact of restriction map capacity.** In Table 3, the *Learned Shared Maps* variant (which removes the data-dependent LoRA modulation) appears to fail on the Maze task, achieving only an 8.9% solved rate. We hypothesized that this collapse was not due to the lack of LoRA modulation, but rather an insufficient channel capacity in the base

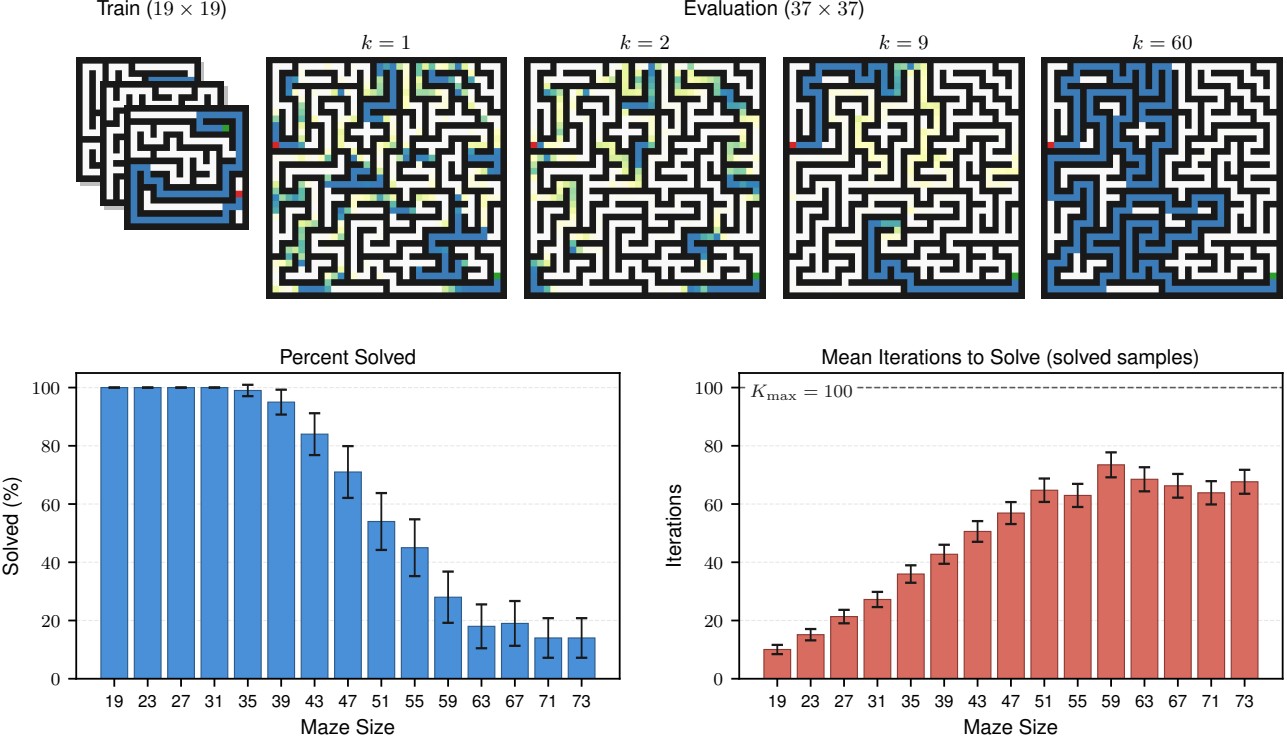

*Figure 6.* **Size generalization in maze pathfinding. Train:** the model is trained only on $19 \times 19$ mazes. **Qualitative:** aggregated path-belief heatmaps on a $37 \times 37$ test maze as a function of the number of ADMM iterations $K$, illustrating how additional coordination steps sharpen a globally consistent shortest-path prediction. **Quantitative:** solved rate (left) and mean iterations-to-solve on solved instances (right) over maze sizes $n \in \{19, 23, \dots, 73\}$, with a maximum inference budget of $K_{\max} = 100$ iterations (dashed line). Error bars indicate variability across evaluation runs.

model configuration ($d_v = 10, d_e = 5$). Without the ability to modulate restriction maps based on local context (LoRA), the encoder must learn a single static linear projection capable of handling all possible topological configurations for a given edge direction. This significantly increases the required expressivity of the communication channel. To test this, we performed a sweep over larger vertex stalk and edge stalk dimensions (Table 5). The results confirm that capacity was indeed the bottleneck: by increasing $d_v$ to 256 and $d_e$ to 32, the static shared maps achieve a 97.7% solved rate, nearly recovering the performance of the LoRA-augmented baseline (99.8%). This demonstrates that LoRA acts as a parameter-efficient compression mechanism, allowing high performance with significantly smaller stalk dimensions.

Furthermore, the ratio between the vertex stalk dimension ($d_v$) and the edge stalk dimension ($d_e$) proves critical: performance collapses when $d_v \leq d_e$ (e.g., $d_v = 32, d_e = 64$ yields 0.2%). Effective coordination requires $d_v$ to be significantly larger than $d_e$, forcing the restriction maps to compress information into a shared agreement space rather than simply passing high-dimensional noise. While increasing $d_e$ beyond 32 yields diminishing returns or performance degradation, increasing $d_v$ consistently improves performance, suggesting that a rich local representation is necessary to fa-

*Table 5.* Ablation on vertex stalk ($d_v$) and edge stalk ($d_e$) dimensions for the **Learned Shared Maps** variant (no LoRA) on the Maze task. While the baseline configuration ($d_v = 10, d_e = 5$) yielded only 8.9% solved rate, scaling up the dimensions allows the static shared maps to recover near-perfect performance (97.7%), comparable to the LoRA-augmented model.

| $d_v$ | $d_e$ | | | | |
|---|---|---|---|---|---|
| | 5 | 16 | 32 | 64 | 128 |
| 10 | 8.9% | 0.0% | 0.0% | 0.0% | 0.0% |
| 16 | 2.2% | 0.0% | 0.0% | 0.0% | 0.0% |
| 32 | 68.2% | 38.1% | 61.1% | 0.2% | 0.0% |
| 64 | 93.6% | 90.8% | 61.1% | 0.2% | 0.0% |
| 128 | 96.6% | 92.8% | 93.3% | 84.6% | 10.6% |
| 256 | 97.8% | 97.1% | 97.7% | 95.3% | 94.7% |

cilitate the simpler, lower-dimensional agreement enforced by the sheaf.

## F. Visualization of Coordination Dynamics

We visualize the optimization process on a single maze example: intermediate predictions and per-agent decision-space trajectories.

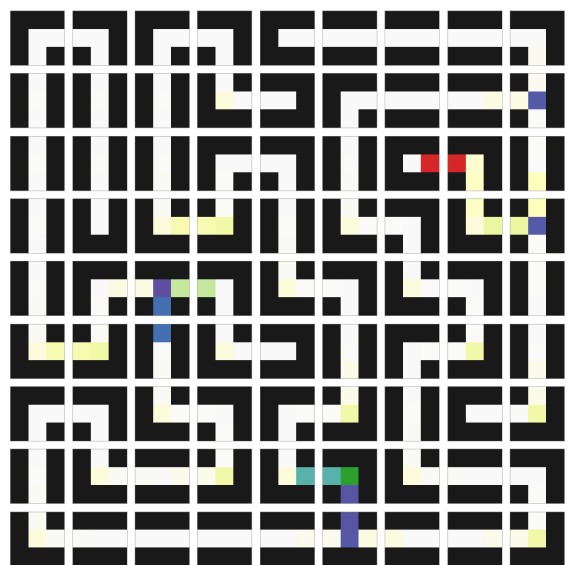

*Figure 7.* **Local views and initial predictions.** A visualization of the inputs and initial states for all 81 agents. The colored overlay indicates the agent's initial prediction confidence prior to any communication. The path predictions are fragmented and lack global connectivity.

### F.1. Local Views and Initial Predictions

To ground the analysis of the coordination dynamics, we first visualize (Figure 7) the local views and the initial outputs of the encoder $\text{Enc}_\theta(\mathbf{d}_i)$ before any consensus constraints are applied. Each cell illustrates exactly what a specific agent $i$ sees: a limited $3 \times 3$ local patch $\mathbf{d}_i$ centered on its location. Additionally, we overlay the agent's initial belief on top of the view.

### F.2. Emergence of Global Consistency

Figure 8 visualizes the forward pass of the unrolled architecture. Initially, predictions exhibit high entropy as agents cannot distinguish the correct path from dead ends. As iterations progress, the sheaf constraints enforce consistency. Locally plausible but globally disconnected segments get progressively pruned until only the shortest remains.

### F.3. Micro-Dynamics of Coordination

The bottom row of Figure 8 plots the negotiation between local preference ($\mathbf{x}_i^k$, blue) and consensus ($\mathbf{z}_i^k$, red) for all 81 agents, revealing the specific mechanics of the ADMM layer. We observe distinct behaviors depending on topological context. Agents located along straight corridors or near dead ends (e.g., agents 10, 16) display nearly identical, straight trajectories. Their local objective $f_i$ aligns quickly with the consensus, requiring little negotiation. Agents at decision points, corners, or near the start/goal (e.g., agent 77 near the start) show nonlinear trajectories, reflecting the system's initial uncertainty regarding the solution direction before

global information is propagated.

## G. Comparison to Recurrent MPNNs

Sheaf-ADMM and recurrent message-passing neural networks (MPNNs) (Gilmer et al., 2017; Li et al., 2016) share a common high-level structure, alternating between communication on the agent graph and per-agent updates. The key differences are twofold. First, Sheaf-ADMM uses update rules with an explicit optimization-derived form, rather than generic learned message and update maps. Second, it separates local proposals, coordination variables, and disagreement memory into distinct and interpretable states ($\mathbf{x}, \mathbf{z}, \mathbf{u}$), whereas a recurrent MPNN typically propagates a single hidden state. These distinctions culminate in substantially different inductive biases.

**Recurrent MPNN.** A recurrent MPNN maintains a single hidden state $\mathbf{h}_i^t \in \mathbb{R}^{d_h}$ at each agent $i$, initialized by an encoder, $\mathbf{h}_i^0 = \text{Enc}_\theta(\mathbf{d}_i)$, where $\text{Enc}_\theta$ is a shared neural network applied to the local view $\mathbf{d}_i$. Each iteration consists of two operations: a message-passing step and an agent update. Messages are computed on edges by a learned message function $\phi$, aggregated at each agent, and then used to update the agent state through a learned recurrent block $\psi$ (e.g., a gated recurrent unit):

$$\mathbf{m}_{j \to i}^t = \phi\big(\mathbf{h}_j^t, \mathbf{h}_i^t\big), \tag{29}$$

$$\bar{\mathbf{m}}_i^t = \text{Agg}_{j \in \mathcal{N}(i)}\, \mathbf{m}_{j \to i}^t, \tag{30}$$

$$\mathbf{h}_i^{t+1} = \psi\big(\mathbf{h}_i^t, \bar{\mathbf{m}}_i^t\big). \tag{31}$$

Both $\phi$ and $\psi$ are learned functions applied at every iteration; depending on the variant, $\phi$ may be realized by an MLP or by a linear map with distinct parameters per direction.

**Structural parallel.** Sheaf-ADMM also performs two distinct operations per outer iteration $k$: an inter-agent communication step (the $\mathbf{z}$-update in (4)) and a per-agent state update (the $\mathbf{x}$-update in (3)). The $\mathbf{z}$-update is the analog of the MPNN's message-passing step, and the $\mathbf{x}$-update is the analog of its agent update.

In the soft-consensus setting, each ADMM iteration approximates

$$\mathbf{z}^{k+1} = \underset{\mathbf{z}}{\arg\min}\ \frac{\gamma}{2}\,\mathbf{z}^\top \mathbf{L}_\mathcal{F}\,\mathbf{z} + \frac{\rho}{2}\big\|\mathbf{z} - \mathbf{v}^k\big\|^2, \\ \mathbf{v}^k := \mathbf{x}^{k+1} + \mathbf{u}^k. \tag{32}$$

by running $T$ gradient-descent (sheaf diffusion) steps from $\mathbf{z}^{(0)} = \mathbf{v}^k$:

$$\mathbf{z}^{(t+1)} = \mathbf{z}^{(t)} - \eta\Big(\gamma\,\mathbf{L}_\mathcal{F}\,\mathbf{z}^{(t)} + \rho\big(\mathbf{z}^{(t)} - \mathbf{v}^k\big)\Big). \tag{33}$$

Because $\mathbf{L}_\mathcal{F}$ only couples neighboring agents, the product $\mathbf{L}_\mathcal{F}\mathbf{z}^{(t)}$ can be computed locally: each edge forms a

disagreement term, and each agent sums the contributions from its incident edges. For an edge $e = (i, j)$, define the disagreement in the edge stalk $\mathbb{R}^{d_e}$ and its contribution to agent $i$:

$$\mathbf{q}_e^{(t)} := \mathbf{F}_{i \to e} \mathbf{z}_i^{(t)} - \mathbf{F}_{j \to e} \mathbf{z}_j^{(t)},$$
$$\mathbf{m}_{e \to i}^{(t)} := -\mathbf{F}_{i \to e}^\top \mathbf{q}_e^{(t)}. \tag{34}$$

One diffusion step can then be written as a local message-passing rule,

$$\mathbf{z}_i^{(t+1)} = \mathbf{z}_i^{(t)} + \eta \gamma \sum_{e \ni i} \mathbf{m}_{e \to i}^{(t)} - \eta \rho (\mathbf{z}_i^{(t)} - \mathbf{v}_i^k). \tag{35}$$

Here $\mathbf{m}_{e \to i}^{(t)}$ plays the role of an incoming message, and $\sum_{e \ni i} \mathbf{m}_{e \to i}^{(t)}$ is the analog of the aggregated message in an MPNN. Compared side by side with the MPNN message rule:

$$\text{MPNN:} \quad \mathbf{m}_{j \to i}^t = \phi(\mathbf{h}_j^t, \mathbf{h}_i^t), \tag{36}$$
$$\text{Sheaf-ADMM:} \quad \mathbf{m}_{e \to i}^{(t)} = -\mathbf{F}_{i \to e}^\top (\mathbf{F}_{i \to e} \mathbf{z}_i^{(t)} - \mathbf{F}_{j \to e} \mathbf{z}_j^{(t)}). \tag{37}$$

This is a more constrained parameterization than an arbitrary learned message map, with a correspondingly stronger inductive bias. The update in (33) is a gradient step for the communication objective in (32). After $T$ diffusion steps, the resulting communication operator has the form $\mathbf{z}^{(T)} = p_T(\mathbf{L}_\mathcal{F}) \mathbf{v}^k$ for a polynomial $p_T$, so it acts as a low-pass filter on $\mathbf{L}_\mathcal{F}$. It also has a well-defined target: projection onto $\ker(\mathbf{F})$ in the hard-consensus limit, or the exact soft update $\rho(\rho \mathbf{I} + \gamma \mathbf{L}_\mathcal{F})^{-1} \mathbf{v}^k$ in the soft-consensus case. A generic recurrent MPNN has no comparable built-in target.

Recent work studies GNN message passing through the propagation dynamics it induces, including vanishing gradients, over-smoothing, and over-squashing (Arroyo et al., 2026; 2025). In our setting, the communication step is more explicit because ADMM isolates it as a separate $\mathbf{z}$-subproblem; in the current linear-quadratic formulation, this makes its behavior more directly analyzable in terms of the spectrum and geometry of $\mathbf{L}_\mathcal{F}$.

Turning to the node update, the Sheaf-ADMM analog of the MPNN recurrent block is the $\mathbf{x}$-update (3). Rather than applying a generic learned recurrent map, this local step is determined by the choice of objective $f_i$: some choices admit closed-form updates, while more general choices, including QPs, require an inner solve. In simple diagonal cases, those closed forms can be as explicit as elementwise division, soft-thresholding, or clipping, so the $\mathbf{x}$-step can take the form of a coordinatewise nonlinearity (Appendix A).

## H. Hyperparameters

All models share an AdamW optimizer with a 200-step linear warmup to a constant learning rate, gradient-norm clipping at 1.0, batch size 128, and an exponential moving average of parameters (decay 0.999) used at evaluation. Matrix multiplications run at full `float32` precision (JAX matmul precision `highest`). Each reported number is a mean over three seeds $\{42, 123, 456\}$. The remaining hyperparameters were selected per task; Table 6 lists the configuration that produces each task's reported result. Ablation rows (Table 3) vary a single one of these axes from the corresponding default.

Table 6. Per-task hyperparameters for the reported configurations. Settings shared across tasks are given in the text; the vertex and edge stalk dimensions $(d_v, d_e)$ and restriction-map sharing are in Section 5.3.

| | MNIST | Maze | Sudoku |
|---|---|---|---|
| Learning rate | $1 \times 10^{-3}$ | $3 \times 10^{-4}$ | $1.7 \times 10^{-3}$ |
| Weight decay | $10^{-7}$ | $10^{-6}$ | $10^{-7}$ |
| Epochs | 20 | 50 | 10 |
| $K$ (train) | 20 | 40 | 20 |
| $K$ (eval) | 100 | 100 | 50 |
| $\rho$ initialization | 0.12 | 0.25 | 0.25 |
| Loss window $w$ | 2 | 4 | 2 |

The three tasks use different specializations of the inner solvers. The $\mathbf{x}$-update applies the diagonal proximal solver of Appendix A: a closed-form $\ell_1$/box step on Maze, and a 50-step accelerated proximal-gradient inner solve for the Lasso objective on MNIST and the non-negativity-constrained objective on Sudoku. The $\mathbf{z}$-update uses the hard consensus constraint $\mathbf{Fz} = \mathbf{0}$ on MNIST and the soft penalty $\frac{\gamma}{2} \|\mathbf{Fz}\|^2$ (Maze $\gamma = 5$, Sudoku $\gamma = 2$) on the others. Both run only $T = 5$ conjugate-gradient steps, so the hard variant approximates the projection onto $\ker(\mathbf{F})$ rather than computing it exactly (Appendix B).

Restriction maps are modulated by a rank-8 (MNIST) or rank-4 (Maze) low-rank update; the reported Sudoku model uses fixed maps, with a rank-4 variant reported alongside it (Table 2). On Maze, the training horizon $K$ is resampled uniformly from $[15, 40]$ at each step. The training loss averages the per-iterate task loss over the final $w$ iterates, rather than supervising the last iterate alone.

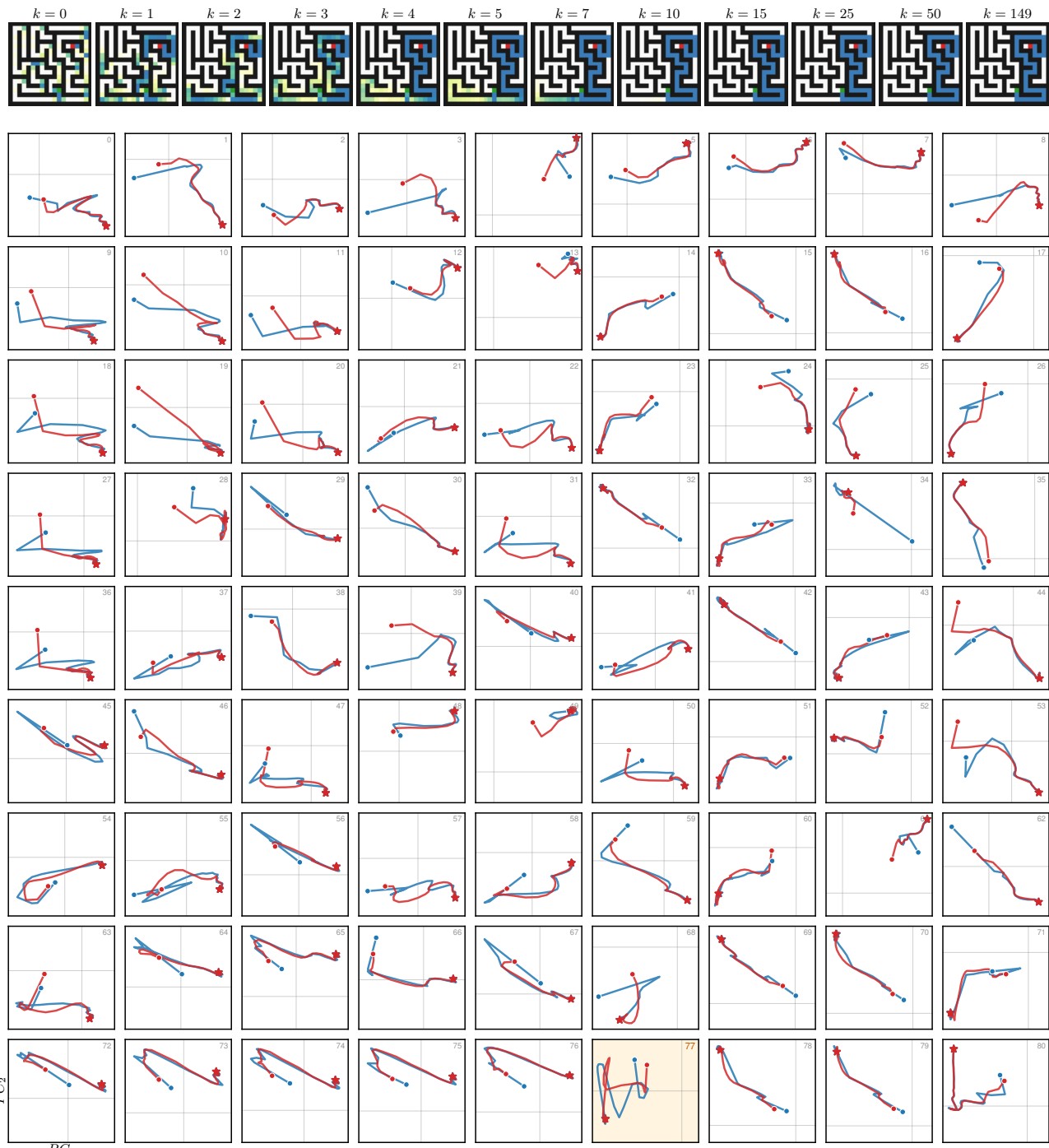

*Figure 8.* **Visualizing coordination mechanisms.** The "tug-of-war" in optimization space. Each subplot in the grid corresponds to the agent at that spatial position. Blue lines track the local decision variable $\mathbf{x}_i^k$, red lines the consensus variable $\mathbf{z}_i^k$, and stars mark the final converged state. Across all 81 agents, the trajectories reveal the negotiation between local preferences and global consensus: agents at decision points (e.g., agent 77 near the green start) exhibit large, nonlinear corrections, while straight, short trajectories indicate regions of low ambiguity (e.g., deep inside walls).

