# OpenReview forum: "Learning Multi-Agent Coordination via Sheaf-ADMM"
_ICML.cc/2026/Conference — ICML 2026 regular_

### Official Review · Reviewer_FEc7 · 2026-03-04

**Soundness:** 3
**Presentation:** 3
**Significance:** 2
**Originality:** 2
**Overall Recommendation:** 4
**Confidence:** 2

**Summary:**

The paper introduces a distributed optimization architecture, in which each "agent" solves a convex subproblem, and they thereby "coordinate" via ADMM using constraints defined by a sheaf, with restriction maps learned from data.

**Compliance With Llm Reviewing Policy:**

Affirmed.

**Final Justification:**

I maintain my overall positive sentiment. While the authors argue for how the currently chosen tasks expose certain mechanistic insights, I still believe broader evaluation would improve the manuscript.

**Key Questions For Authors:**

Could the evaluation suite be expanded to less toy problems, and could comparison to existing methods be considered?

**Limitations:**

Yes

**Strengths And Weaknesses:**

This paper is positioned considerably outside my usual area of research. I have tried to conduct a review of surrounding literature to the best of my ability.

The methodology seems sound. The novelty seems to be in the synthesis of existing ideas in the literature. Specifically, the combination of (1) learning a sheaf, together with (2) using that sheaf to implement the ADMM via sheaf diffusion, (3) demonstrating the efficacy of the combination on local-patches MNIST, mazes, sudoku.

The biggest weakness of the paper in my view is lack of comparison to prior work. How does the authors' method compare against message passing models for e.g. pathfinding? Similar concerns for the sudoku evaluation. Overall the chosen benchmarks seem quite toy and custom, and the lack of evaluation of competing methods makes it difficult to assess the paper's position in the literature and its significance. However, the OOD generalization to larger maze sizes is interesting, and gives some credence for the architecture as proof of concept.

The presentation of the paper is also reasonable, and citations of surrounding literature seem to be fairly comprehensive.

---

> ### Author Rebuttal · Authors · 2026-03-31
>
> > The biggest weakness of the paper in my view is lack of comparison to prior work. How does the authors' method compare against message passing models for e.g. pathfinding? Similar concerns for the sudoku evaluation.
>
> We agree that external baselines were needed. We chose a locality-matched MPNN/GGNN baseline because it preserves the core constraint of our setting: each agent has only a local observation and must coordinate over the same communication graph (Li et al., ICLR 2016; Gilmer et al., ICML 2017). To make this comparison generic rather than bespoke, we used four MPNN variants formed by crossing two capacity controls with two aggregation choices: matched-parameter vs matched-channel, each with either max or mean aggregation. For every task, we kept the message-passing horizon $K$ matched to the Sheaf-ADMM unroll budget and re-optimized the MPNN learning rate. All numbers below are multi-seed (3 seeds, mean±std).
>
> **Sudoku**: Sheaf fixed-RM achieves **84.3 ± 1.6%**, versus **32.2 ± 6.6%** for the best MPNN baseline (matched-parameter, max aggregation), a **+52 point** gap despite using fewer parameters (**1.12M** vs **4.62M**). This is the clearest setting for our method: a non-spatial overlapping consistency problem on a 27-node constraint graph (rows, columns, boxes), where explicit agreement geometry should matter most and where recurrent relational baselines are most relevant (Palm et al., NeurIPS 2018).
>
> **Maze** (21x21), $K=40$, exact solve rate): Sheaf achieves **99.9 ± 0.1 / 98.1 ± 1.2 / 4.5 ± 1.1** on test maze sizes of 21x21 / 2x size / 4x size, versus **99.9 ± 0.1 / 68.3 ± 2.4 / 1.3 ± 1.0** for the best MPNN baseline, again the matched-parameter max-aggregation variant. Both families solve the in-distribution test set, which is consistent with BF/DP-aligned GNN work on graph algorithms and pathfinding (Velickovic et al., ICLR 2020; Dudzik and Velickovic, NeurIPS 2022).
>
> **MNIST**: Sheaf reaches **98.6 ± 0.5%**, the best MPNN **98.9 ± 0.0%**, and a standard CNN **99.1 ± 0.1%**.
>
> We include these new baseline comparisons and a comprehensive discussion of these results in the revised version of the manuscript.
>
> > Overall the chosen benchmarks seem quite toy and custom... Could the evaluation suite be expanded to less toy problems?
>
> We understand the concern that standard MNIST, Sudoku, and Mazes can appear as toy problems. However, our primary motivation for this work is not to compete with monolithic models on standard centralized benchmarks, but rather to explore collective intelligence as a fundamental computational paradigm. We aim to understand the minimal coordination mechanisms required for a network of severely restricted, decentralized agents to achieve global consensus. We selected these environments to create highly controlled, explicitly interpretable testbeds for this specific challenge.
>
> To further push the boundaries of our method and explore less traditional problem structures, we designed several new stress-test tasks during this rebuttal phase, including Permuted MNIST, Permuted Maze, and Bit Majority. These tasks strip away the standard spatial priors to test global and semantic coordination. We detail these new findings comprehensively in our response to **Reviewer RnCg** and update our Limitations section to reflect the boundaries they exposed.
>
> We hope the inclusion of the new MPNN and GNN baselines and the expanded experimental evaluations address your primary concerns.

---

> > ### Author Rebuttal · Reviewer_FEc7 · 2026-04-01
> >
> > My thanks to the authors.
> >
> > The locality-matched MPNN/GGNN baseline inclusion seems like a strong choice for baseline comparison. The lift on sudoku over the baselines seems the most convincing result in the updated paper.
> >
> > My concern is not simply that MNIST/maze/sudoku are small or synthetic. But rather it is that such testbeds may fail to expose problem nuances that specifically arise once coordination problems scale: for example, under much worse conditioning of the consensus operator, more heterogeneous communication, and/or tighter compute budgets under which the same ADMM iteration count no longer suffices. In other words, more modern benchmark-like domains can better test whether the proposed coordination mechanism remains effective outside carefully chosen and controlled settings. With choosing such highly controlled tasks, there is also the concern for a task-selection confound, where the chosen tasks align with the inductive biases of the method. So there are merits for evaluation on standardized benchmarks even outside of benchmark competition per se.
> >
> > I therefore see the current experiments as strong proof-of-concept for the mechanism, but not yet as evidence that the same mechanism will transfer to larger or qualitatively richer coordination regimes. Still, the paper is clearly written, and is now better positioned in the literature. I maintain my score of 4.

---

> > > ### Author Response · Authors · 2026-04-03
> > >
> > > Thank you for the follow-up. We agree with the core assessment: the current experiments are strong proof-of-concept for Sheaf-ADMM, but do not yet demonstrate transfer to larger or qualitatively richer coordination regimes. We want to address the failure modes you raise: **(1)** conditioning, **(2)** heterogeneous communication, and **(3)** tighter compute budgets, as well as **(4)** the task-selection confound, because we believe the structure of Sheaf-ADMM already provides concrete tools for each.
> > >
> > > ## 1. Conditioning
> > >
> > > Conditioning of the consensus operator is a challenge for any local-view method, MPNNs included. In Sheaf-ADMM, the consensus step is formulated as an optimization problem over the sheaf Laplacian $L_{\mathcal{F}}$, and any method that reduces the same objective is a valid $z$-step. This exposes well-studied knobs for improving conditioning -- conjugate gradient (CG) in place of gradient descent, preconditioning, adaptive $\rho$ [2], over-relaxation [1] -- without changing the rest of the architecture. We already see CG dramatically outperform GD+Nesterov on this step (Table 1), and since submission we have found preconditioning, adaptive $\rho$, and over-relaxation each beneficial. We do not claim to have fully validated these knobs at scale, but we view Sheaf-ADMM favorably precisely because it makes conditioning a structured, addressable problem.
> > >
> > > ## 2. Heterogeneous communication
> > >
> > > We agree this is an important axis, and one we have only lightly explored -- heterogeneity is in fact one of the predominant motivations of the sheaf formalism [3,4]. The sheaf structure already accommodates different state dimensions per node and edge-specific agreement relations, so heterogeneous coordination is native to the formalism.
> > >
> > > ## 3. Compute budgets
> > >
> > > The same tools discussed above -- CG, over-relaxation, etc. -- each improve effective coordination per iteration without increasing $K$. We have not yet evaluated under genuinely tight iteration budgets, and that remains open, but much of the classical ADMM literature targets fast, stable convergence in compute-constrained regimes, with techniques that apply directly to Sheaf-ADMM [1].
> > >
> > > ## 4. Task selection
> > >
> > > We take the task choice concern seriously. The paper's goal is to study a coordination mechanism and the properties it exposes: **(1)** toolbox access to the wider ADMM and sheaf-theoretic literature (e.g. the conditioning discussion above), **(2)** robustness under distribution shift, and **(3)** interpretable $x$/$z$/$y$ dynamics (Figures 4, 6) -- rather than to compete on benchmarks. The tasks were chosen to expose the mechanism under controlled conditions.
> > >
> > > For instance, on MNIST, the local-view decomposition is inherently unfavorable compared to a single CNN, yet it exposes properties such as robustness to padding and noise (Table 2) that are not visible through test accuracy alone. The MPNN shares the same view decomposition but does not exhibit the same robustness profile on MNIST, especially under padding and noise (see our response to **Reviewer RnCg**). We hypothesize that the primal-dual structure encourages the encoder to learn representations that serve the **coordination mechanism itself**, not just representations that solve the training distribution. We note that we have equivalent results on CIFAR-10, omitted from the manuscript for space/simplicity, which show the same pattern. We therefore believe the task selection already provides evidence for the properties (1)-(3) above.
> > >
> > > That said, we agree that the controlled settings may be hiding failure modes. We outlined explicit failure conditions in Section 6.3, and we would find it genuinely valuable to demonstrate the method in a regime where those conditions hold -- not to show success, but to characterize the boundary. We are happy to prioritize such a study, or to include additional tasks, in the revision if the reviewer deems it useful. We hope the above discussion better contextualizes our choice of tasks.
> > >
> > > In any event, we wish to thank the reviewer -- the MPNN comparison in particular has substantially improved the manuscript.
> > >
> > > ## References
> > > [1] Boyd et al., "Distributed Optimization and Statistical Learning via the Alternating Direction Method of Multipliers" (2011).
> > >
> > > [2] Xu, Figueiredo, and Goldstein, "Adaptive ADMM with Spectral Penalty Parameter Selection" (2017).
> > >
> > > [3] Curry, *Sheaves, Cosheaves and Applications* (2014).
> > >
> > > [4] Hanks et al., "Distributed Multi-Agent Coordination over Cellular Sheaves" (2025).

---

### Official Review · Reviewer_SoVF · 2026-03-08

**Soundness:** 3
**Presentation:** 3
**Significance:** 3
**Originality:** 3
**Overall Recommendation:** 4
**Confidence:** 2

**Summary:**

This manuscript proposes a differentiable optimization framework named Sheaf-ADMM to address the multi-agent coordination problem. The framework decomposes the input into local views, where each agent utilizes a neural network encoder to solve a local convex subproblem, and coordinates with neighboring agents via the Alternating Direction Method of Multipliers (ADMM) and cellular sheaf constraints. By backpropagating through the unrolled optimization process, the model enables end-to-end joint training of the encoders, decoders, and the sheaf structure. Experiments demonstrate that, even when agents possess only limited local views insufficient to independently solve the task, the framework not only successfully tackles image classification, maze pathfinding, and Sudoku, but also exhibits superior robustness to distribution shifts (e.g., padding and noise) compared to standard CNNs, while endowing the coordination dynamics with direct interpretability.

**Compliance With Llm Reviewing Policy:**

Affirmed.

**Key Questions For Authors:**

1. Could you provide a detailed quantitative comparison of the peak training memory (VRAM) and inference computational complexity (FLOPs) between Sheaf-ADMM (under a typical $K=30$ iterations) and baseline models (such as standard CNNs or GNNs)?
2. May I ask whether the encoder outputs a unified $\Delta F_i$ for agent $i$ that is applied to all its connected edges $e$, or does it output independent low-rank updates for each specific directed edge $(i \to e)$?  If the former, how does it handle the highly heterogeneous coordination demands with different neighbors?
3. If a certain proportion of spurious edges or missing edges is artificially injected into the grid graph or Sudoku constraint graph during training or testing, what does the performance degradation curve of the model look like?

**Limitations:**

yes

**Strengths And Weaknesses:**

**Strengths：**
- 1. End-to-end learning of coordination logic: By employing LoRA to parameterize the sheaf structure inherent in algebraic topology, this manuscript overcomes the constraints of traditional multi-agent systems that depend on manually hard-coded communication protocols.
- 2. Robustness and size generalization: The manuscript makes decisions by relying on the iterative negotiation between local views and global consensus, which not only provides high resilience against data perturbations (e.g., padding, noise) but also enables zero-shot generalization to larger-sized inputs.
- 3. Interpretability：By explicitly decoupling local computation from global communication, the manuscript allows for the direct observation of the dynamic interplay between local preferences and global consensus for each agent, thereby breaking the black-box nature of traditional graph networks.
**Weaknesses：**
- 1. Constrained by iteration budgets and long-range bottlenecks: Long-range coordination strongly depends on the number of unrolled ADMM iterations $K$.  If the task diameter exceeds the coverage range of $K$, it leads to a precipitous drop in performance, whereas blindly increasing $K$ will incur severe memory and computational latency bottlenecks.
- 2. Strong reliance on the local overlap prior: The effectiveness of the architecture is predicated on the assumption that the task can be decomposed into overlapping subproblems. If the task heavily relies on non-local features, or if the manually specified initial communication topology is flawed, this mechanism will completely fail.
- 3. Lack of comparison with external state-of-the-art (SOTA) baselines: In core logical reasoning tasks such as maze pathfinding and Sudoku, the experiments are strictly confined to ablation analyses of internal mechanisms (e.g., LoRA, solvers, and the $K$ value), completely lacking performance comparisons with existing mainstream methods tailored for graph structures or distributed reasoning (e.g., GNNs, MPNNs, or neural algorithmic reasoning models), which leaves the actual advantages of the proposed method over mature existing architectures without sufficient empirical support.

---

> ### Author Rebuttal · Authors · 2026-03-31
>
> > Lack of comparison with external state-of-the-art (SOTA) baselines... completely lacking performance comparisons with existing mainstream methods tailored for graph structures or distributed reasoning (e.g., GNNs, MPNNs, or neural algorithmic reasoning models).
>
> We agree that additional baselines are necessary to anchor our results. Please see our response to **Reviewer FEc7** for a detailed discussion and new empirical comparisons against GNN and MPNN baselines.
>
> > Could you provide a detailed quantitative comparison of the peak training memory (VRAM) and inference computational complexity (FLOPs)?
>
> Because Sheaf-ADMM unrolls an iterative solver, analyzing its computational footprint is crucial. We leverage gradient checkpointing over the ADMM steps ($K$) to bound memory usage and fit all experiments on a single H100 80GB GPU.
>
> In our representative Maze pathfinding setting (**181,861** parameters), the graph contains 81 nodes and 272 edges. Using $K=30$ iterations and 5 inner CG steps, the model requires approximately **65.0 MFLOPs** per example at inference. This breaks down into the ADMM loop (**~36.0 MFLOPs**), Encoder (**~24.2 MFLOPs**), and Decoder (**~4.9 MFLOPs**). Peak training VRAM was measured at **61 GB**. Compared to our baselines, Sheaf-ADMM is roughly **1.5x** more expensive at inference. A matched-parameter MPNN baseline requires approximately **43 MFLOPs** per example at $K=30$.
>
> For heavier configurations:
> - **MNIST** (**655,405** params): scales from **237.5 MFLOPs** at $K=10$ to **502.1 MFLOPs** at $K=30$.
> - **Sudoku** (**678,914** params): scales from **387.1 MFLOPs** at $K=10$ to **895.7 MFLOPs** at $K=30$.
>
> While Sheaf-ADMM currently requires more compute than standard monolithic baselines, this implementation serves primarily as a proof-of-concept for decentralized architectures. In practice, the footprint could be reduced by standard ADMM enhancements such as warm-starting and reusing primal/dual variables across iterations.
>
> > If spurious or missing edges are artificially injected... what does the performance degradation curve look like?
>
> To address the impact of a partially flawed communication graph, we artificially injected varying proportions of rewired edges (randomly removing valid edges and adding spurious ones) into the test graphs for MNIST. We found the degradation is smooth and gradual:
> - **Baseline (0% rewired)**: **99.03%** Accuracy (Agreement: **99.79%**)
> - **10% rewired**: **75.67%** Accuracy (Agreement: **93.77%**)
> - **20% rewired**: **72.98%** Accuracy (Agreement: **91.99%**)
> - **30% rewired**: **54.57%** Accuracy (Agreement: **88.19%**)
>
> We include these results and a discussion of this performance degradation curve in the revised version of the manuscript.
>
> We also evaluated eval-time edge drop on the maze test graph and compared exact solve rate against the new MPNN baselines (3 seeds, mean±std). At 0.5/1/3/5% drop, Sheaf achieves **82.5±22.4 / 79.9±2.4 / 31.8±0.5 / 5.6±0.8**. The matched-parameter MPNN with max aggregation reaches **74.0±2.2 / 53.1±1.9 / 22.7±2.7 / 7.0±1.4**, while the matched-parameter mean-aggregation MPNN reaches **87.5±0.4 / 66.5±1.1 / 38.9±1.3 / 16.1±0.7**. The matched-dimension baselines are weaker overall: max gives **47.6±20.8 / 35.3±12.5 / 16.0±2.0 / 6.2±2.1**, and mean gives **68.5±1.9 / 48.3±1.1 / 17.9±3.6 / 4.1±2.3**. Overall, the degradation is again gradual rather than catastrophic. Sheaf is stronger than the max-aggregation baselines across light-to-moderate corruption, while the matched-parameter mean-aggregation MPNN is the most robust under heavier corruption.
>
> > Does the encoder output a unified $\Delta F_i$ for agent $i$ applied to all edges, or independent low-rank updates for each directed edge?
>
> Per-node, per-direction LoRA factors. They are direction-conditioned: on grids, neighbors in the same cardinal direction share modulation; on sudoku, the slot indexes constraint type. This captures heterogeneous coordination demands while keeping parameterization efficient.
>
> > Long-range coordination strongly depends on $K$... blindly increasing $K$ will incur severe memory and computational latency bottlenecks.
>
> On our maze task (2x size OOD): **74%** at $K=40$, **99.6%** at $K=100$. While inference-time $K$ can be increased freely, training-time large $K$ remains a real cost.
>
> Thus, we agree that blindly scaling the unrolled horizon $K$ creates memory and latency bottlenecks. To scale to substantially longer-range tasks efficiently, a natural architectural extension is to implement hierarchical decompositions or multi-scale agents (as noted in our Future Directions). By routing information through a hierarchy, we can exponentially reduce the graph's effective diameter, allowing rapid propagation of global consistency without requiring a deep unrolling horizon.

---

### Official Review · Reviewer_pYbm · 2026-03-13

**Soundness:** 3
**Presentation:** 3
**Significance:** 2
**Originality:** 3
**Overall Recommendation:** 4
**Confidence:** 3

**Summary:**

This paper introduces Sheaf-ADMM, a differentiable multi-agent optimization framework that combines Cellular Sheaf Theory with unrolled ADMM to solve structured prediction tasks through local coordination.

**Compliance With Llm Reviewing Policy:**

Affirmed.

**Final Justification:**

My concerns are addressed during the rebuttal. I increased my score.

**Key Questions For Authors:**

1. The current benchmarks (MNIST, Mazes) have a very natural spatial decomposition. How would the Sheaf-ADMM framework handle tasks where the 'agreement' is not spatial but semantic?

**Limitations:**

Yes

**Strengths And Weaknesses:**

**Strength**:
1. The paper presents an elegant integration of Cellular Sheaf Theory and unrolled ADMM, which is an interesting and solid idea.

**Weakness**:
1. While effective on structured tasks like MNIST and mazes, the reliance on convex local subproblems may limit the framework's applicability to high-dimensional, non-convex domains about modern tasks.
2. The experiments lack comparison with modern SOTA baselines like transformer based methods.
3. The iterative nature of unrolled ADMM significantly increases the computational cost

---

> ### Author Rebuttal · Authors · 2026-03-31
>
> > While effective on structured tasks like MNIST and mazes, the reliance on convex local subproblems may limit the framework's applicability to high-dimensional, non-convex domains about modern tasks.
>
> We agree this is a potential limitation, but we think it is less severe than it may first appear. The full model is still nonconvex: the encoder and decoder are neural networks, so convexity is a property of the latent solver, not a restriction on the model's overall expressivity. The main benefit of convexity is that it gives the coordination layer the cleanest ADMM semantics and standard convergence guarantees. At the same time, nonconvex ADMM is a substantial literature rather than a non-starter (e.g., Hong et al., SIAM J. Optim. 2016; Wang et al., J. Sci. Comput. 2019), where ADMM remains a useful heuristic even with nonconvex subproblems. So our choice to focus on convex subproblems in this paper is a deliberate design choice, not a strict requirement of the framework.
>
> > The experiments lack comparison with modern SOTA baselines like transformer based methods.
>
> We completely agree that additional baselines help better contextualize our method's performance. Please see our response to **Reviewer FEc7** for a detailed discussion and new empirical comparisons against standard SOTA baselines for distributed reasoning on graphs.
>
> > The iterative nature of unrolled ADMM significantly increases the computational cost
>
> It is true that the iterative coordination mechanism fundamentally requires more compute per forward pass than a monolithic network. We leverage gradient checkpointing over the ADMM steps to strictly bound memory usage, allowing all experiments to comfortably fit on a single 80GB GPU. Please see our response to **Reviewer SoVF** for a detailed, quantitative breakdown of the FLOPs and peak VRAM.
>
> We emphasize that this method is not currently intended to replace highly optimized monolithic models (like transformers) in standard centralized settings. Rather, we are exploring the minimal mechanisms required for decentralized groups of restricted agents to coordinate.
>
> > The current benchmarks (MNIST, Mazes) have a very natural spatial decomposition. How would the Sheaf-ADMM framework handle tasks where the 'agreement' is not spatial but semantic?
>
> This is a great question. In fact, our framework already handles semantic agreement successfully. In the Sudoku task, agents correspond to logical constraint groups (rows, columns, boxes) rather than spatial patches. The agreement here is purely semantic, resolving discrete logical conflicts.
>
> To further prove this, please see our response to **Reviewer RnCg**. We ran additional experiments on the Permuted MNIST where spatial adjacency is completely destroyed. The model remained resilient, demonstrating that the sheaf constraints can successfully learn to aggregate distributed, semantic feature evidence even through a randomized topology.
>
> We hope this addresses your questions regarding the framework's mechanics, baseline comparisons, and semantic capabilities.

---

> > ### Author Rebuttal · Reviewer_pYbm · 2026-04-04
> >
> > Thanks for the reply. I think my concerns are addressed by the rebuttal. By adding more strong baselines, I think the result is now solid. I will increase my rating.

---

> > > ### Author Response · Authors · 2026-04-06
> > >
> > > Thank you for the positive assessment. We are glad the additional baselines resolved your concerns, and we appreciate the feedback that pushed us to better contextualize the method against existing architectures.

---

### Official Review · Reviewer_RnCg · 2026-03-23

**Soundness:** 2
**Presentation:** 3
**Significance:** 2
**Originality:** 2
**Overall Recommendation:** 4
**Confidence:** 3

**Summary:**

The paper proposes Sheaf-ADMM, a differentiable multi-agent coordination framework where an input is split into local views, each handled by an agent solving a learned convex subproblem, with inter-agent consistency enforced through ADMM and a cellular sheaf that specifies what neighboring agents must agree on. The system is trained end-to-end by unrolling ADMM, and is evaluated on MNIST classification, maze pathfinding, and Sudoku.

**Compliance With Llm Reviewing Policy:**

Affirmed.

**Final Justification:**

The rebuttal addresses several of my main concerns, especially by adding stronger GNN/MPNN baselines and by probing topology sensitivity more directly. These additions make the empirical case more credible and clarify where the method works and where it breaks. I am adjusting my score to 4.

**Key Questions For Authors:**

How would Sheaf-ADMM perform on tasks where the target depends on genuinely nonlocal structure rather than local compatibility constraints?
How sensitive is the method to the choice of agent graph and overlap pattern?
Can the framework scale to problems that require substantially longer-range coordination than the current tasks?

**Limitations:**

The evaluation is concentrated on tasks that are highly aligned with the method’s inductive bias.
The method may be fragile when the agent graph or overlap structure is poorly chosen.
The paper shows that increasing the number of ADMM iterations improves performance up to a point, but also saturation or degradation beyond roughly 10–30 iterations. This suggests that tasks requiring much longer coordination horizons may be difficult to handle in practice.

**Strengths And Weaknesses:**

Strength:
- Clear optimization formulation: local agent objectives, a consensus constraint, sheaf restriction maps, and an unrolled ADMM solver with trainable encoder/decoder and learned restriction maps.
- Each agent carries explicit primal, consensus, and dual variables, and the ADMM loop exposes these dynamics directly.
- The tasks are diverse.
- Results are strong on structured tasks.
Weaknesses:
- Lack baseline comparisons against strong alternatives.
- MNIST robustness results compare Sheaf-ADMM only against a standard CNN

---

> ### Author Rebuttal · Authors · 2026-03-31
>
> > Lack baseline comparisons against strong alternatives.
>
> We agree that additional baselines help better contextualize our method's advantages. We have run new baseline comparisons against GNNs and MPNNs. Please see our detailed response to **Reviewer FEc7** for the complete baseline comparison.
>
> > MNIST robustness results compare Sheaf-ADMM only against a standard CNN.
>
> We have now included our MPNN baselines in the MNIST robustness comparison. Under input perturbations, all three models start near parity on clean data (Sheaf **98.5%**, CNN **99.3%**, MPNN **98.7%**). On padding, Sheaf-ADMM retains a clear advantage: at pad-16, Sheaf holds **86.3%** while the CNN drops to **11.4%** and the MPNN to **15.8%**. On pixel drop, Sheaf and MPNN are comparable (**97.0%** vs **97.6%** at 5% drop; **69.1%** vs **66.5%** at 30% drop), both outperforming the CNN (**94.4%** and **45.5%** respectively).
>
> > How would Sheaf-ADMM perform on tasks where the target depends on genuinely nonlocal structure rather than local compatibility constraints? How sensitive is the method to the choice of agent graph and overlap pattern? The method may be fragile when the agent graph or overlap structure is poorly chosen.
>
> We thank you for these insightful questions. We grouped them together as they prompted us to design a new suite of stress-test experiments to evaluate Sheaf-ADMM's reliance on communication topology and its performance on non-local spatial tasks.
>
> **Sensitivity to Overlap Prior**: We evaluated the method's sensitivity by sweeping through varying strides (1, 2, 3) and patch sizes (3x3, 5x5). We found the method is highly robust to these variations. Across all variants, MNIST maintained **97–99%** test accuracy, and Maze Pathfinding maintained a **99–100%** solved rate.
>
> **Flawed Topology and Non-Local Tasks**: To test the model's reliance on local compatibility, without introducing entirely new datasets, we evaluated the model on inputs subjected to a fixed, random permutation. Permuting the input acts as a direct proxy for applying a completely flawed, randomized communication topology to a normal input.
>
> - **Permuted MNIST**: By permuting pixels, local patches no longer contain spatially adjacent data. Surprisingly, the model remained resilient, achieving validation accuracy within the margin of error of the baseline (though with slightly higher training loss: **0.06** vs **0.03**). This demonstrates that for tasks where "agreement" is semantic (aggregating distributed feature evidence), the sheaf constraints can still learn to route information effectively through a flawed spatial topology.
>
> - **Global Counting (Bit Majority)**: We evaluated a random binary image classification task with the goal of classifying bit majority. This requires a proxy of global counting, making the task genuinely non-local. The method manages to solve it to a significant extent (val acc: **92.63%**).
>
> - **Permuted Maze**: When we applied the same permutation to the Maze task, the model failed completely. Unlike MNIST, maze solving strictly depends on the continuous adjacency of the grid. When the communication topology is disconnected from the input's spatial structure, the model is unable to propagate the necessary continuous path information.
>
> We appreciate you pushing us to explore these boundaries, and we have updated our Limitations section to explicitly discuss these failure modes on tasks lacking local compatibility.
>
> > Can the framework scale to problems that require substantially longer-range coordination than the current tasks?
>
> As you correctly point out, increasing the coordination horizon relies heavily on the number of unrolled ADMM iterations ($K$). As the problem size scales, relying solely on a larger $K$ to propagate local consistency across a massive graph becomes computationally bottlenecked.
>
> To scale to substantially longer-range tasks without requiring an infinitely deep unrolling horizon, a natural extension is to implement hierarchical decompositions or multi-scale agents (as noted in our Future Directions). By introducing an agent hierarchy, we exponentially reduce the effective diameter of the communication graph while preserving the strict spatial adjacency required for tasks like pathfinding.
>
> We hope that the inclusion of the new GNN/MPNN baselines and the additional experimental results on sensitivity to graph topology directly address your core concerns.

---

> > ### Author Rebuttal · Reviewer_RnCg · 2026-04-04
> >
> > The rebuttal addresses several of my main concerns, especially by adding stronger GNN/MPNN baselines and by probing topology sensitivity more directly. These additions make the empirical case more credible and clarify where the method works and where it breaks.
> > I am adjusting my score to 4.

---

> > > ### Author Response · Authors · 2026-04-06
> > >
> > > Thank you for the acknowledgement. We are glad the MPNN baselines and topology sensitivity experiments strengthened the empirical case. We appreciate the questions that prompted the new stress-test experiments, which we believe meaningfully improved the paper.

---

### Decision · Program_Chairs · 2026-04-30

**Decision:**

Accept (regular)

**Comment:**

This paper proposes Sheaf-ADMM, a differentiable multi-agent coordination framework. The method decomposes the input into overlapping local views, with each agent solving a parameterized local convex subproblem, and enforces consistency through ADMM and cellular sheaf constraints. The paper evaluates the framework on MNIST, maze pathfinding, and Sudoku, demonstrating its ability to recover correct global solutions under limited local information, while also reporting robustness under distribution shift and providing interpretable analyses of the coordination dynamics.

The paper’s strengths lie in its clear formulation, which combines sheaf structure, ADMM-based coordination, and end-to-end learning into a coherent and interpretable framework. The rebuttal further strengthened the empirical case by adding comparisons with MPNN/GGNN baselines, topology-sensitivity analyses, computational cost measurements, and degradation studies under graph corruption. However, the current version still has several limitations. First, the experiments remain concentrated on controlled tasks that align closely with the method’s inductive bias, and therefore do not yet show that the mechanism can reliably generalize to larger-scale, more heterogeneous, or longer-range coordination problems. Second, although stronger external baselines were added, the overall evaluation scope is still limited and does not yet fully establish the method’s advantage over existing graph-based or distributed coordination approaches. Therefore, I recommend weak accept.